# Community composition shapes microbial-specific phenotypes in a cystic fibrosis polymicrobial model system

Fabrice Jean-Pierre[1]*, Thomas H Hampton[1], Daniel Schultz[1], Deborah A Hogan[1], Marie-Christine Groleau[2], Eric Déziel[2], George A O'Toole[1]*

[1]Department of Microbiology and Immunology, Geisel School of Medicine at Dartmouth, Hanover, United States; [2]Centre Armand-Frappier Santé Biotechnologie, Institut National de la Recherche Scientifique, Laval, Canada

*For correspondence:
jeanpierre.fabrice@gmail.com
(FJ-P);
georgeo@dartmouth.edu
(GAO'T)

**Competing interest:** The authors declare that no competing interests exist.

**Abstract** Interspecies interactions can drive the emergence of unexpected microbial phenotypes that are not observed when studying monocultures. The cystic fibrosis (CF) lung consists of a complex environment where microbes, living as polymicrobial biofilm-like communities, are associated with negative clinical outcomes for persons with CF (pwCF). However, the current lack of in vitro models integrating the microbial diversity observed in the CF airway hampers our understanding of why polymicrobial communities are recalcitrant to therapy in this disease. Here, integrating computational approaches informed by clinical data, we built a mixed community of clinical relevance to the CF lung composed of *Pseudomonas aeruginosa*, *Staphylococcus aureus*, *Streptococcus sanguinis*, and *Prevotella melaninogenica*. We developed and validated this model biofilm community with multiple isolates of these four genera. When challenged with tobramycin, a front-line antimicrobial used to treat pwCF, the microorganisms in the polymicrobial community show altered sensitivity to this antibiotic compared to monospecies biofilms. We observed that wild-type *P. aeruginosa* is sensitized to tobramycin in a mixed community versus monoculture, and this observation holds across a range of community relative abundances. We also report that LasR loss-of-function, a variant frequently detected in the CF airway, drives tolerance of *P. aeruginosa* to tobramycin specifically in the mixed community. Our data suggest that the molecular basis of this community-specific recalcitrance to tobramycin for the *P. aeruginosa lasR* mutant is increased production of phenazines. Our work supports the importance of studying a clinically relevant model of polymicrobial biofilms to understand community-specific traits relevant to infections.

## Editor's evaluation

This article establishes a new model of the multispecies microbiome of infections of the cystic fibrosis airway and demonstrates its utility and reproducibility for testing the effectiveness of antibiotics and understanding how different bacteria interact. It shows that antibiotic susceptibility of the dominant pathogen *P. aeruginosa* changes in this mixed community relative to when grown alone, and indicates that a commonly-arising mutant of *P. aeruginosa* alters this sensitivity. This report is significant as a community resource and for its discovery of one reason why antimicrobial treatments can fail in polymicrobial infections.

## Introduction

The disconnect between in vitro antimicrobial susceptibility profiles and clinical response poses a significant threat to the eradication of mixed microbial communities observed in human infections

(*Lebeaux et al., 2014*). That is, while minimal inhibitory concentration (MIC) assays are widely used to guide clinical intervention by determining the antimicrobial susceptibility profiles of pathogenic species, such approaches often fail to resolve chronic, polymicrobial infections (*Orazi and O'Toole, 2019*; *Waters et al., 2019*). Studies published several decades ago documented that microbe-microbe interactions can drive shifts in antimicrobial susceptibility profiles (*Linn and Szabo, 1975*; *Shahidi and Ellner, 1969*). Thus, antibiotics effective at killing a single microorganism in vitro may be ineffective against the same microbe when grown in the context of a polymicrobial infection. More recent studies provide some insight as to the mechanisms driving resistance and tolerance (also referred to as recalcitrance or resilience) toward antimicrobials in the context of mixed-species infections (*Beaudoin et al., 2017*; *Bottery et al., 2022*; *O'Brien et al., 2022*; *Orazi et al., 2020*; *Orazi and O'Toole, 2017*; *Orazi et al., 2019*). These mechanisms include (but are not limited to) the production of metabolites altering microbial physiology, horizontal gene transfer of genetic material, and passive protection by the production of shared 'public goods', such as β-lactamases (reviewed in *Orazi and O'Toole, 2019*; *Vandeplassche et al., 2019*).

Over the last decade, both culture-based and culture-independent studies have indicated that chronic cystic fibrosis (CF) lung infections are characterized by the presence of numerous microbial taxa (*O'Sullivan and Freedman, 2009*; *O'Toole, 2018*; *Rogers et al., 2003*; *Surette, 2014*). The clinical relevance of such mixed communities is highlighted by the observations that co-occurrence of *Pseudomonas aeruginosa* and *Staphylococcus aureus* alters the antimicrobial susceptibility of the latter organism and worsens CF lung disease (*Orazi et al., 2020*; *Orazi and O'Toole, 2017*; *Orazi et al., 2019*; *Limoli et al., 2016*). Furthermore, antibiotics effective against pathogens such as *P. aeruginosa* in classic in vitro MIC assays show limited clinical efficacy, as indicated by the inability of tobramycin to clear this microorganism from the CF airway (*Heirali et al., 2020*; *Nelson et al., 2020*; *Ramsey et al., 1999*). Indeed, a recent report recommends abandoning MIC testing for microorganisms isolated from CF airway infections as their predictive value for a positive treatment outcome is not supported (*Waters et al., 2019*).

The pathogenesis of mixed-species infections in the CF airway is still poorly understood, but it is now appreciated that distinct community types can impact clinical outcomes (*Hampton et al., 2021*; *Widder et al., 2022*). However, one of the current missing links allowing for the translation of microbiome-informed studies back to the clinic is the establishment of in vitro mixed-community models that can be used to understand community function(s) of CF pathogens (*Jean-Pierre et al., 2021*). That is, although some CF-relevant model communities have been proposed (*Orazi et al., 2020*; *Orazi and O'Toole, 2017*; *Orazi et al., 2019*; *Flynn et al., 2020*; *Flynn et al., 2016*; *Vandeplassche et al., 2020*), they (i) do not entirely reflect the polymicrobial nature of the CF airway and/or (ii) do not utilize in vitro conditions recapitulating the nutritional milieu and/or biofilm-like growth observed in the CF lung. Thus, there is a pressing need for the development of clinically informed, in vitro polymicrobial communities to probe the molecular mechanism(s) governing microbial interactions, particularly in regard to the responsiveness to antimicrobial agents.

The aims of this study were fourfold: (i) to leverage large, existing microbiome data sets and their associated clinical metadata to inform and develop a clinically relevant and tractable in vitro model of the polymicrobial communities found in the CF airway, (ii) to identify community-specific phenotypes, (iii) to test this in vitro model system against the most common front-line CF antimicrobial, tobramycin (*Cystic Fibrosis Foundation Patient Registry, 2021*), and (iv) to provide a mechanistic understanding whereby community composition can impact tobramycin responsiveness.

## Results

### Using computational approaches to identify community types found in the CF airway

We sought to exploit available 16S rRNA gene amplicon library sequencing data and associated clinical metadata to identify a representative set of microbial communities found in persons with CF (pwCF) that we could model in vitro. A recent study from our team using k-means to cluster 16S rRNA gene amplicon sequencing relative abundance values and another machine learning approach (e.g. the gap statistic) identified five microbial community clusters as the most parsimonious number (*Hampton et al., 2021*). This same study revealed, following the analysis of a large cross-sectional

16S rRNA gene sequence data set and associated metadata, that the five community types explained 24% of variability in lung function, twice as high as any other factor or group of factors previously identified (*Hampton et al., 2021*). Thus, two different approaches, using clinical data, identified five similar community types.

Of these five communities, two were dominated by a single microbial taxa: *P. aeruginosa* (designated the 'Pa.D' community) and *Streptococcus* (Strep.D); these communities included 73/167 (~43%) of the analyzed samples (*Hampton et al., 2021*). Several *P. aeruginosa*-focused CF studies make it clear that specifically targeting this pathogen does not translate to positive clinical outcomes in pwCF (*Heirali et al., 2020*; *Nelson et al., 2020*; *Acosta et al., 2021*; *Heirali et al., 2017*), and there has already been a substantial focus on the study of *P. aeruginosa* in monoculture in the context of CF (*Malhotra et al., 2019*; *Rossi et al., 2021*). Thus, we deemed that in vitro modeling of a *P. aeruginosa*-dominated community would not allow us to probe unknown/novel factors driving chronic CF lung disease. Furthermore, while the presence of *Streptococcus* spp. may influence CF airway health, the clinical relevance of this microbe in CF is still a matter of active research, and this genus is associated with both worsened and improved clinical outcomes (*Scott and O'Toole, 2019*). We identified a third cluster designated Oth.D, for 'other'; this cluster was composed of samples from pwCF whose lung microbiota were dominated by less-common, individual genera including *Stenotrophomonas*, *Burkholderia*, and *Achromobacter*. Based on these factors, we decided to focus on two community types identified in the study cited above (*Hampton et al., 2021*), which are composed of various microbial taxa (Pa.M1/Pa.M2; which we refer to as 'mixed'). The Pa.M1 and Pa.M2 mixed communities represented ~34% of the analyzed samples in this study (*Hampton et al., 2021*).

As previously reported (*Henson et al., 2019*), we performed metabolic flux analyses of the various community types to identify the top predicted exchanged metabolites driving the abundances of *P. aeruginosa*, *S. aureus*, *Streptococcus sanguinis*, and *Prevotella melaninogenica* by using their known relative 16S rRNA gene abundance detected in the Pa.M1/Pa.M2, Pa.D, and Strep.D communities (*Figure 1—figure supplement 1A*). Using this approach, we noted multiple similarities between the predicted metabolic fluxes of Pa.M1 and Pa.M2 that distinguished these communities from the *Pseudomonas*- and *Streptococcus*-dominated communities, for example, the fluxes of ethanol, lactate, and acetate (*Figure 1—figure supplement 1A*). Furthermore, as the predicted metabolic fluxes between Pa.M1 and Pa.M2 were similar, we considered these two mixed communities as functionally similar and decided to focus on the development of a single 'mixed' community model.

Next, we sought to identify a limited number of community members to model as a tractable in vitro mixed community. We noted that 10 microbial taxa achieved relatively high abundance and prevalence across the five identified community types (*Hampton et al., 2021*; *Figure 1A*). From among these taxa, we decided to focus on *P. aeruginosa*, *S. aureus*, *Streptococcus* spp., and *Prevotella* spp. as members of our model mixed community for the following reasons: (i) *P. aeruginosa*, *S. aureus*, *Streptococcus* spp., and *Prevotella* spp. are important in shaping health outcomes in pwCF based on studies using culture-dependent and -independent approaches (*Malhotra et al., 2019*; *Filkins et al., 2012*; *Lamoureux et al., 2021*; *Sibley et al., 2008*; *Thornton et al., 2022*; *Ulrich et al., 2010*; *Waters and LiPuma, 2020*; *Zemanick et al., 2013*). (ii) Additional studies provide evidence of the potential for two or more of these microorganisms found together in the CF airway to lead to worsened clinical outcomes (*Limoli et al., 2016*; *Limoli and Hoffman, 2019*; *Sherrard et al., 2016*). (iii) Imaging studies of sputum have shown evidence for a subset of these microbes being in close physical proximity (*DePas et al., 2016*; *Jennings et al., 2021*). (iv) Assessing the fraction of 16S rRNA reads across the data sets analyzed by Hampton, O'Toole, and colleagues (*Hampton et al., 2021*), *Pseudomonas*, *Staphylococcus*, *Streptococcus*, and *Prevotella* were among the top 10 microbial taxa detected in the CF airway with relative abundances of 35, 3, 20, and 13% and prevalence rates of 70, 50, 90, and 80%, respectively (*Hampton et al., 2021*; *Figure 1A*, red box). (v) In 103 of the 167 (~62%) patient samples analyzed by Hampton, O'Toole, and colleagues, >70% of 16S rRNA gene reads were assigned to *Pseudomonas*, *Staphylococcus*, *Streptococcus*, and *Prevotella* (*Figure 1B*); another 30 samples contained at least 50% of the reads assigned to these four genera. Including the next most abundant and prevalent genera such as *Burkholderia* or *Achromobacter* as a fifth member of the mixed community would only marginally increase (by ~6%) the number of patient samples covered (*Figure 1—figure supplement 1B and C*). Furthermore, the average abundance of *Burkholderia* was skewed by a relatively small number of pwCF in our cohort that were dominated by this pathogen

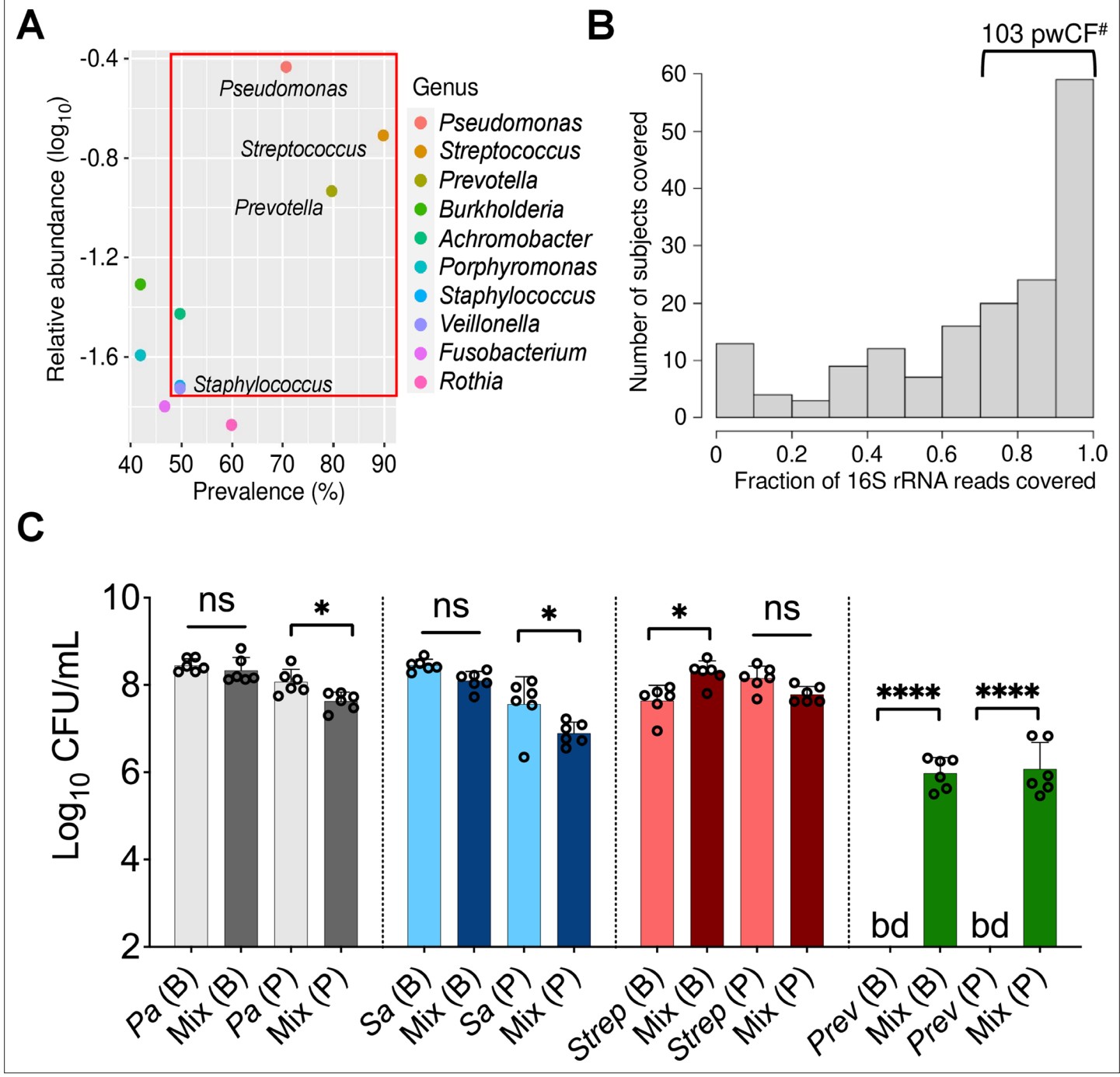

**Figure 1.** Leveraging clinical microbiome data sets with computational analyses to identify communities and community members to model in vitro. (**A**) Relative 16S rRNA gene abundance and prevalence of the top 10 cystic fibrosis (CF) lung pathogens in the 167 persons with CF (pwCF) data set used as the basis for developing the in vitro mixed community, as reported by *Hampton et al., 2021*. (**B**) Number of unique samples for which >70% of 16S rRNA reads are associated with the combined presence of *Pseudomonas*, *Staphylococcus*, *Streptococcus*, and *Prevotella*. #Indicates the number of samples (103) that meet this criterion from the total sample size of 167 pwCF. (**C**) Colony forming units (CFUs) counts of each microbial member grown as a monoculture and in a mixed community (Mix) for biofilm (**B**) and planktonic (**P**) fractions. CFUs were performed by plating on medium selective for the growth of each microorganism. Each data point presented in a column represents the average from at least three technical replicates performed at least on three different days (n=6). Statistical analysis was performed using ordinary one-way ANOVA and Tukey's multiple comparisons posttest with *, p<0.05; ****, p<0.0001, ns = non-significant. Error bars represent SD. Pa = *Pseudomonas aeruginosa*, Sa = *Staphylococcus aureus*, Strep = *Streptococcus sanguinis*, Prev = *Prevotella melaninogenica,* and bd = below detection.

The online version of this article includes the following source data and figure supplement(s) for figure 1:

*Figure 1 continued on next page*

*Figure 1 continued*

**Figure supplement 1.** Leveraging clinical microbiome data sets with computational analyses to identify communities and community members to model in vitro.

**Figure supplement 2.** *Prevotella melaninogenica* does not grow as monoculture biofilms in artificial sputum medium (ASM) in anoxic conditions.

**Figure supplement 3.** Productivity of *P. aeruginosa*, *S. aureus*, *S. sanguinis*, and *P. melaninogenica* grown as monospecies or mixed planktonic/biofilm communities.

**Figure supplement 3—source data 1.** Raw data for productivity calculation of *P. aeruginosa*, *S. aureus*, *S. sanguinis*, and *P. melaninogenica* grown in various conditions.

**Figure supplement 4.** Microbial composition range of in vivo CF mixed communities compared with the in vitro model.

**Figure supplement 4—source data 1.** Raw data for in vitro relative abundance of *P. aeruginosa*, *S. aureus*, *S. sanguinis*, and *P. melaninogenica* grown in various ratios in mixed communities.

**Figure supplement 5.** Testing additional laboratory and clinical strains in the in vitro polymicrobial community model.

**Figure supplement 6.** Fourteen days co-culture experiment of community members grown in a planktonic and biofilm mixed communities.

---

(*Hampton et al., 2021*). (vi) Finally, work published by our group using metabolic modeling indicates that *P. aeruginosa*, *S. aureus*, *S. sanguinis*, and *P. melaninogenica* are top contributors of cross-fed metabolites in communities detected in the CF lung (*Henson et al., 2019*). That is, the abundance of these four genera in the CF airway could be explained by predicted metabolic cross-feeding among these four microorganisms. Thus, we settled on a mixed community composed of *P. aeruginosa*, *S. aureus*, *Streptococcus* spp., and *Prevotella* spp., which represents a common 'pulmotype' detected in ~34% of airway infections for pwCF, as the basis for the development of an in vitro model system.

## Implementing a mixed in vitro model system to probe community function

Based on the data outlined in the previous section, we focused on four bacteria: *P. aeruginosa*, *S. aureus*, *S. sanguinis*, and *P. melaninogenica*. We cultivated laboratory strains of *P. aeruginosa* PA14, *S. aureus* Newman, *S. sanguinis* SK36, and *P. melaninogenica* ATCC25845 as monocultures and mixed communities in anoxic conditions, which reflects the CF airway environment (*Worlitzsch et al., 2002*), and using artificial sputum medium (ASM), which mimics the nutritional conditions of the CF lung (*Palmer et al., 2005*; *Turner et al., 2015*). We could quantify the respective viable counts from both biofilm and planktonic populations for each of these four microorganisms by plating on selective media, as shown in *Figure 1C* and described below.

The *P. aeruginosa* biofilm population did not show statistically significant differences in endpoint colony forming unit (CFU) counts at 24 hr when cultivated as either a monoculture or in a mixed community (*Figure 1C*, gray bars). However, a modest (~0.5 log) but statistically significant decrease was detected for *P. aeruginosa* grown planktonically in the mixed community (*Figure 1C*, gray bars). While not negatively impacted in a mixed community versus monoculture in a biofilm, *S. aureus* viable counts were reduced in a planktonic community versus when cultivated as monoculture (*Figure 1C*, blue bars). A statistically significant increase in biofilm CFU counts was observed for *S. sanguinis* in the mixed community (*Figure 1C*, red bars). However, no differences were detected for planktonic cells of this microorganism in the presence of microbial partners (*Figure 1C*, red bars). Interestingly, we could not detect *P. melaninogenica* in monoculture, a finding predicted in our previous metabolic modeling study (*Henson et al., 2019*), but ~6 $\log_{10}$ CFU/mL of this microorganism could be detected when cultivated in the presence of other microbial partners in both biofilm and planktonic fractions (*Figure 1C*, green bars).

The impact of residual oxygen negatively influencing the growth of *P. melaninogenica* in monoculture was ruled out by performing these experiments using an anoxic environmental chamber (*Figure 1—figure supplement 2*). That is, we did not detect CFU counts for either planktonic or biofilm populations of *P. melaninogenica* when grown in ASM in the anoxic environmental chamber, but as a positive control, significant growth was detected when using a medium shown previously to support growth of this microbe (*Jang et al., 2016*; *Prevotella* growth medium, or PGM; *Figure 1—figure supplement 2*).

Calculating the theoretical productivity (or yield) of each microorganism grown as planktonic or biofilm monospecies or as mixed communities, as defined using a reported method (*Poltak and*

*Cooper, 2011*), and comparing these values to the experimentally determined populations revealed several interesting findings (*Figure 1—figure supplement 3*). First, the yield of monospecies biofilm populations of *P. aeruginosa* and *S. aureus* was higher than planktonically grown cells, while the opposite was observed for *S. sanguinis* (*Figure 1—figure supplement 3A*). Furthermore, the observed productivity in the mixed planktonic community (denoted as 'Mix (P) – Observed') was lower than the theoretical productivity (i.e. assuming additivity of the CFU/mL yield counts of each species in the community, denoted as 'Mix (P) – Theoretical'), suggesting antagonism among the different species grown in these conditions (*Figure 1—figure supplement 3A*). Interestingly, examining the yield of the observed mixed biofilm and planktonic populations (both denoted as 'Mix (P/B) – Observed') revealed that the productivity of the mixed biofilm is higher than that of the planktonic community (*Figure 1—figure supplement 3A*). There was no statistically significant difference in the yield of observed and theoretical mixed biofilm communities (*Figure 1—figure supplement 3A*). However, taking a closer look at the productivity of each species in the observed mixed community revealed that there is a trend wherein *P. aeruginosa* and *S. aureus* have lower productivity in the mixed biofilm versus monoculture (p=0.67; *Figure 1—figure supplement 3B*). On the other hand, *S. sanguinis* showed a statistically significant increase in productivity in a mixed biofilm community compared to the monospecies biofilm (p=0.03; *Figure 1—figure supplement 3B*). Overall, these data indicate a complex relationship among the microbes in the community, which varies depending on whether they are growing planktonically or in a biofilm.

The in vitro mixed biofilm community under the model growth conditions resulted in a composition within the range observed for the in vivo M1/M2 mixed communities previously reported (*Hampton et al., 2021*; *Figure 1—figure supplement 4A*). Moreover, by maintaining *P. aeruginosa* at the same starting concentration and shifting the inoculum of *S. aureus*, *S. sanguinis*, and *P. melaninogenica*, varying mixed community compositions reflecting the microbial populations in the CF lung were observed (*Figure 1—figure supplement 4—source data 1*).

While the data shown in *Figure 1C* pertains to commonly used laboratory strains, similar observations were made for multiple other strains and/or CF clinical isolates of these four microbial genera (*Figure 1—figure supplement 5*). Finally, we assessed whether this model system could maintain these microbial populations over time by replacing the medium every 24 hr. We stably detected each member of the mixed community for up to 2 weeks (*Figure 1—figure supplement 6*). Taken together, our data shows that we can model an in vitro mixed community reflective of the polymicrobial infection in the CF airway found in the airway of ~34% of pwCF and that the microorganisms in this model display community-specific growth phenotypes.

## Polymicrobial context shifts tobramycin sensitivity of CF pathogens

Using the newly developed model described above, we sought to test the hypothesis that the susceptibility of CF pathogens to tobramycin would shift when treated in the context of a mixed community. Tobramycin was selected as it is one of the most heavily prescribed antimicrobials in the CF clinic *Cystic Fibrosis Foundation Patient Registry, 2021*; we used clinically relevant concentrations in all studies (*Ruddy et al., 2013*).

Initially focusing on biofilm communities, which are more recalcitrant to antimicrobial therapy (*Mah and O'Toole, 2001*), we observed that tobramycin treatment of wild-type (WT) *P. aeruginosa* PA14 grown in the mixed community resulted in an unexpected reduction in the number of viable cells compared to monoculture (*Figure 2A*). Furthermore, no detectable counts of *P. aeruginosa* in the planktonic phase of the mixed community were observed (*Figure 2A*). Notably, inoculating the community with a 1000× fold less CFU of *S. aureus*, *S. sanguinis*, and *P. melaninogenica* than *P. aeruginosa* rescued the tolerance phenotype of this microbe to tobramycin (*Figure 2—figure supplement 1* labeled in red). These data demonstrate increased killing of *P. aeruginosa* by tobramycin in a polymicrobial environment can occur over a range of abundances of other microbial partners, but there may be an eventual lower limit to the presence of other partners to observe community-specific phenotypes.

We expanded our analysis beyond the PA14 laboratory strain of *P. aeruginosa*. Of the six *P. aeruginosa* CF clinical isolates tested (including mucoid and non-mucoid strains), two strains displayed a similar phenotype to strain PA14 (*Figure 2—figure supplement 2*: strains SMC1596, AMT0101-1-2, labeled in red), one strain showed equal sensitivity in monoculture and the polymicrobial community

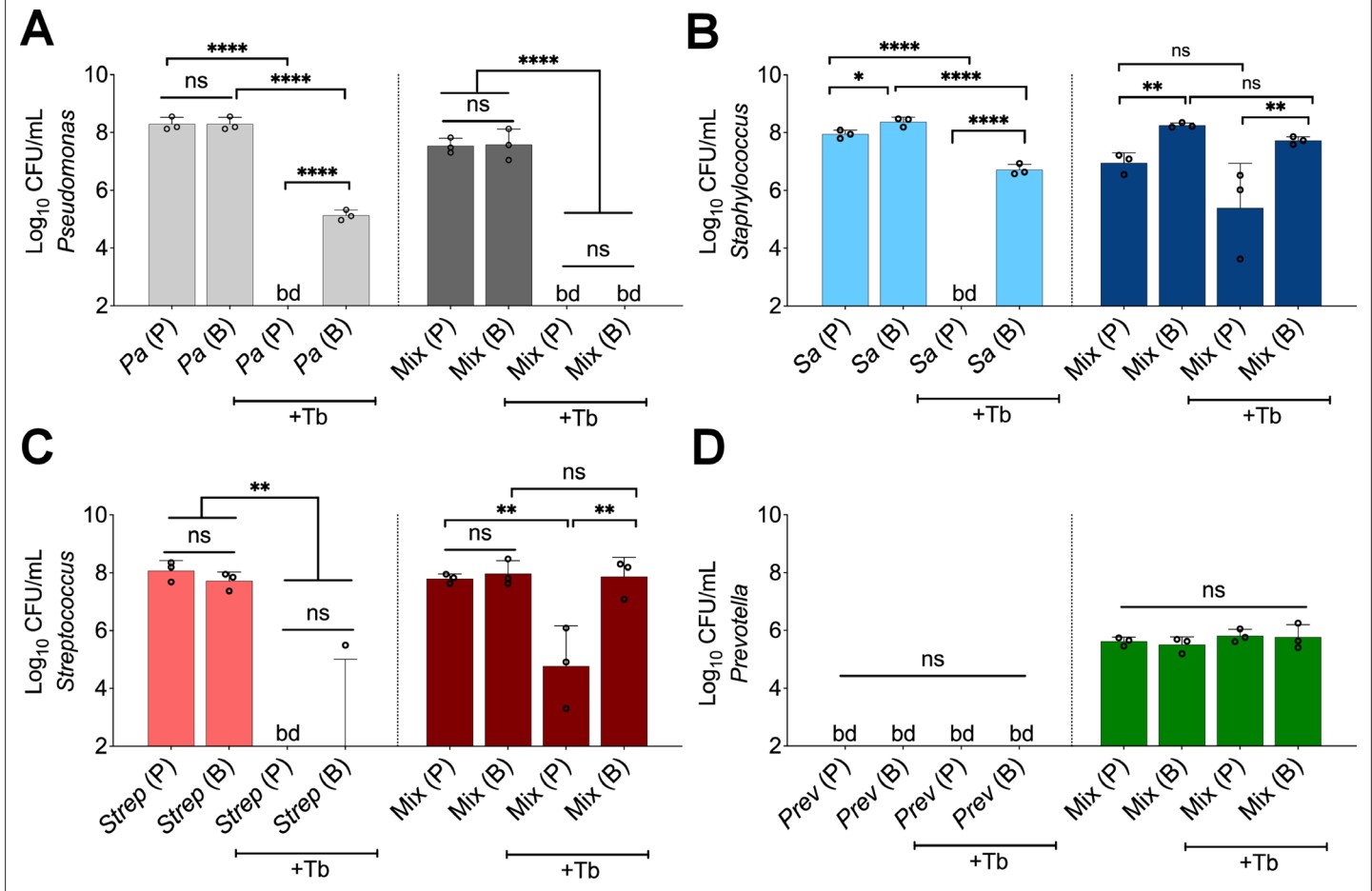

**Figure 2.** Polymicrobial context shifts tobramycin sensitivity of CF pathogens. Colony forming units of planktonic (**P**) and biofilm (**B**) populations of (**A**) *P. aeruginosa* (Pa), (**B**) *S. aureus* (Sa), (**C**) *S. sanguinis* (Strep), and (**D**) *P. melaninogenica* (Prev) grown as monocultures or mixed communities (Mix) and challenged or not with 100 µg/mL of tobramycin (+Tb). Each data point presented in a column represents the average from at least three technical replicates performed at least on three different days (n=3). Statistical analysis was performed using ordinary one-way ANOVA and Tukey's multiple comparisons posttest with *, $p<0.05$; **, $p<0.01$; ****, $p<0.0001$, ns = non-significant, and bd = below detection. Error bars represent SD.

The online version of this article includes the following figure supplement(s) for figure 2:

**Figure supplement 1.** Microbial partners increase the killing of *P. aeruginosa* exposed to tobramycin in a mixed community over a wide range of population sizes.

**Figure supplement 2.** Drug sensitivity of *P. aeruginosa* clinical strains grown in a mixed community challenged with tobramycin.

**Figure supplement 3.** Shifted sensitivity of *P. aeruginosa* PA14 grown in a mixed community using various laboratory strains and clinical isolates treated with tobramycin.

**Figure supplement 4.** Recalcitrance of *S. aureus* biofilms grown in a mixed community composed of various strains and treated with tobramycin.

**Figure supplement 5.** Recalcitrance of *Streptococcus* spp. biofilms grown in a mixed community composed of various strains and treated with tobramycin.

**Figure supplement 6.** Recalcitrance of *Prevotella* spp. biofilms grown in a mixed community composed of various strains and treated with tobramycin.

**Figure supplement 7.** Polymicrobial context shifts tobramycin sensitivity of CF pathogens in fully anoxic conditions.

(*Figure 2—figure supplement 2*: strain SMC1587), while one clinical isolate displayed increased tolerance in a polymicrobial environment (*Figure 2—figure supplement 2*: strain SMC1595, labeled in blue). A similar enhancement of tobramycin-mediated killing was observed when *P. aeruginosa* PA14 was co-cultivated with multiple *Streptococcus* and *Prevotella* species (*Figure 2—figure supplement 3*). Interestingly, co-cultivating *P. aeruginosa* PA14 with *S. aureus* strain USA300 did not impact tobramycin sensitivity of *P. aeruginosa* in the mixed community while strains Newman and JE2 did

alter sensitivity (*Figure 2—figure supplement 3C*), indicating that only a subset of strains of *S. aureus* might contribute to the community-specific shift in *P. aeruginosa* sensitivity to tobramycin.

We noted several other changes in tobramycin sensitivity in the context of the mixed community compared to monoculture. Most *S. aureus* strains and *Streptococcus* spp. showed decreased tobramycin susceptibility when cultivated in the mixed community in both planktonic and biofilm populations (*Figure 2B*, *Figure 2C*, *Figure 2—figure supplement 4*, *Figure 2—figure supplement 5*). *S. aureus* strain USA300 showed high level tolerance to tobramycin in monoculture but was not further protected from tobramycin in the mixed community (*Figure 2—figure supplement 4A*). We could not determine shifts in *Prevotella* spp. sensitivity to tobramycin in monoculture as this microorganism cannot be cultivated under these conditions in the absence of the other community members (*Figure 2*, *Figure 2—figure supplement 6*). Growing and replenishing the pre-formed biofilm communities with fresh ASM supplemented or not with tobramycin using an anoxic environmental chamber resulted in similar phenotypes for all tested microorganisms (*Figure 2—figure supplement 7*), indicating that the use of the GasPak system provides a robust anoxic environment.

Community-specific protection of bacterial members was unlikely driven by microbial-based inactivation of the drug as the minimal bactericidal concentration (MBC) value remained similar to ASM control when *P. aeruginosa* was treated with tobramycin that was pre-incubated with microbial supernatants for 24 hr (*Supplementary file 1*). This conclusion was supported by the observation that tobramycin treatment still effectively killed some of the microbes in the community. Taken together, we demonstrated that growth in the mixed community shifts the antimicrobial sensitivity of multiple bacteria compared to growth in monoculture in anoxic conditions.

## LasR loss of function increases tobramycin tolerance of *P. aeruginosa* in the mixed community

The unexpected increase in sensitivity of *P. aeruginosa* when challenged with tobramycin in the mixed community (*Figure 2A*) prompted us to reconcile our in vitro observations with what is observed in the clinic. That is, culture-independent microbiome studies do not indicate any appreciable changes in the abundance of *P. aeruginosa* in the airways of pwCF post-treatment with tobramycin (*Heirali et al., 2020*; *Nelson et al., 2020*; *Ramsey et al., 1999*).

*P. aeruginosa* undergoes genetic adaptation in the airway of pwCF (*Smith et al., 2006*), and loss-of-function mutations in the gene coding for the key intercellular communication regulator, LasR (*Lee and Zhang, 2015*), are frequently observed in the airway of pwCF and other environments (*Feltner et al., 2016*; *Groleau et al., 2022*; *Hoffman et al., 2009*; *Mould et al., 2022*; *Robitaille et al., 2020*). Strains lacking LasR function have also been associated with worsened lung function (*Hoffman et al., 2009*), in vitro growth advantage in low oxygen conditions (*Clay et al., 2020*), increased production of virulence factors through intraspecies interactions (*Mould et al., 2020*), and greater tolerance to front-line CF drugs (*Hoffman et al., 2010*). We thus hypothesized that loss of LasR function might result in a different response toward tobramycin treatment in the community.

To test this hypothesis, we replaced WT *P. aeruginosa* PA14 in the mixed population with an isogenic Δ*lasR* mutant. The absence of LasR function resulted in increased tolerance of this *P. aeruginosa* mutant to tobramycin in the mixed community (*Figure 3A*). This phenotype was complemented by restoring a WT *lasR* allele at the native locus (*Figure 3A*). Importantly, the Δ*lasR* mutant showed a similar degree of sensitivity to tobramycin as the WT when these strains were grown in monoculture (*Figure 3A*). The recalcitrance of the Δ*lasR* mutant was observed to be biofilm-specific as planktonic cells grown in a mixed community displayed equal sensitivity to the drug when compared to monoculture (*Figure 3B*). Endpoint CFU counts of other microbial members in the mixed community were not impacted by the inactivation of the *lasR* gene with or without tobramycin treatment (*Figure 3—figure supplement 1*). Inactivation of tobramycin by the Δ*lasR* mutant was unlikely as the *P. aeruginosa* MBC remained similar to ASM control when WT *P. aeruginosa* was treated with tobramycin pre-incubated in the supernatant of the Δ*lasR* mutant (*Supplementary file 1*).

To determine whether the altered tolerance of LasR loss-of-function is not specific to the *P. aeruginosa* PA14 strain, we tested a chronic CF clinical isolate, NC-AMT0101-1-1, defective for LasR activity (LasR⁻; *Smith et al., 2006*). The strain displayed higher tolerance to tobramycin in the mixed community than its parent isolate (NC-AMT0101-1-2; LasR⁺; *Figure 3C*). Both NC-AMT0101-1-1 and NC-AMT0101-1-2 isolates were equally sensitive to tobramycin in monoculture, suggesting a

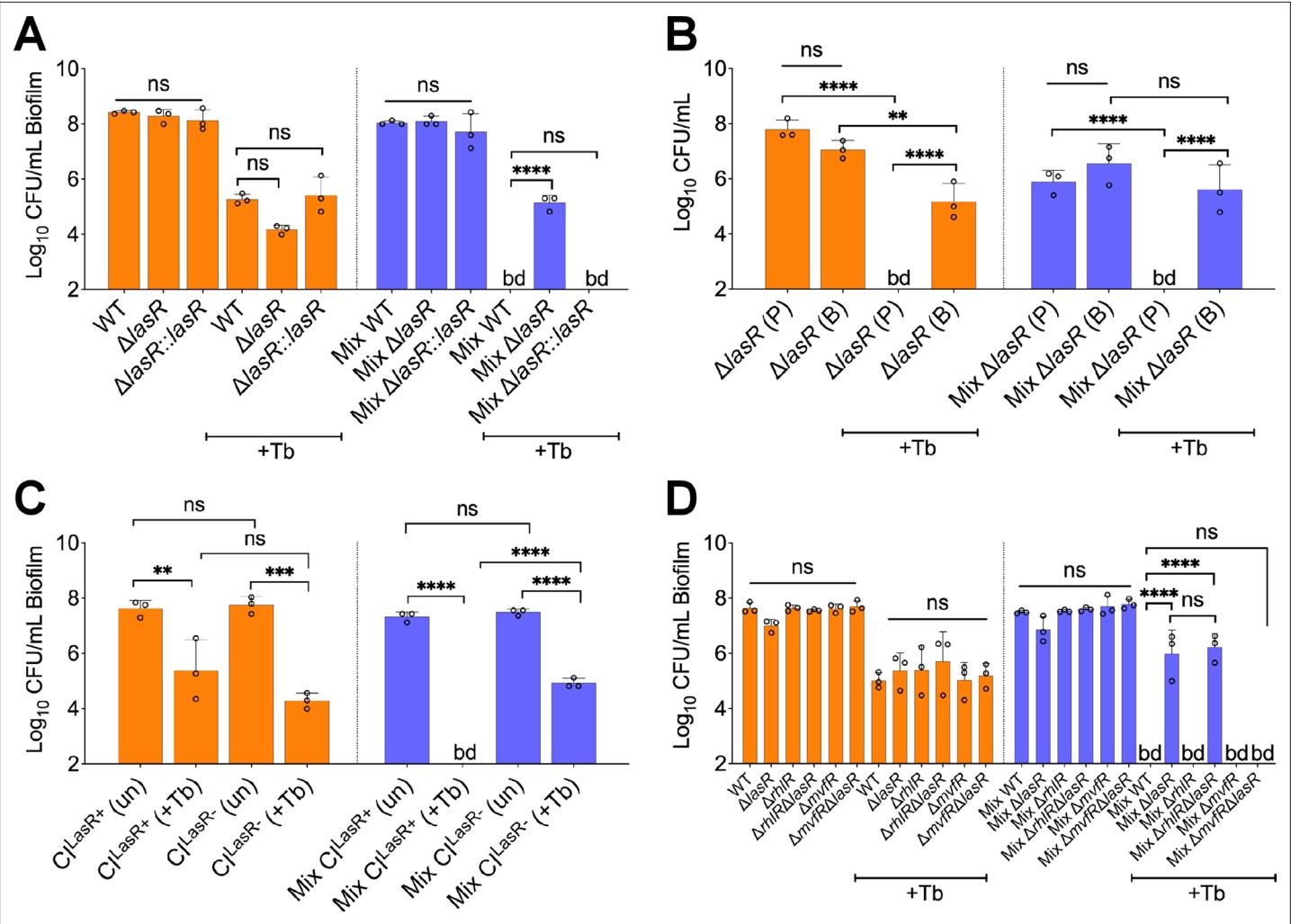

**Figure 3.** LasR loss-of-function drives biofilm-specific tobramycin tolerance in a mixed community. Colony forming units of (**A**) *P. aeruginosa* strain PA14 (wild-type [WT]), isogenic Δ*lasR* mutant, and the complemented strain (Δ*lasR::lasR*), (**B**) planktonic and biofilm Δ*lasR* mutant cells, (**C**) LasR-defective (NC-AMT0101-1-1; LasR−) and LasR+ (NC-AMT0101-1-2) clinical isolates (CIs) and (**D**) *P. aeruginosa* quorum sensing regulator mutants grown as monocultures and mixed communities (Mix) and challenged or not with 100 µg/mL of tobramycin (+Tb). Each data point presented in a column represents the average from at least three technical replicates performed at least on three different days (n=3). Statistical analysis was done using ordinary one-way ANOVA and Tukey's multiple comparisons posttest with \*\*, p<0.01; \*\*\*, p<0.001; \*\*\*\*, p<0.0001. Error bars represent SD. bd = below detection and un = untreated.

The online version of this article includes the following figure supplement(s) for figure 3:

**Figure supplement 1.** Loss of *P. aeruginosa* LasR function does not alter the viability of the other microbes in the mixed community compared to growth with wild-type (WT) *P. aeruginosa*.

**Figure supplement 2.** LasR-specific phenotypic tests of *P. aeruginosa* CF clinical isolate SMC1595.

**Figure supplement 3.** Tolerance of Δ*lasR* mutant in a mixed community: a role for the MvfR/PQS regulatory system.

community-specific phenotype for LasR loss-of-function in the model (*Figure 3C*). These results further prompted us to test the hypothesis that the clinical isolate SMC1595 lacked LasR function as this strain is more tolerant to tobramycin treatment in a mixed community (*Figure 2—figure supplement 2*). Indeed, both low protease activity and reduced 3-oxo-$C_{12}$ homoserine lactone (3-oxo-$C_{12}$-HSL) production, features associated with LasR loss-of-function variants (*Feltner et al., 2016*; *Groleau et al., 2022*; *Mould et al., 2022*), were observed in this strain when compared to controls (*Figure 3—figure supplement 2*).

The LasR quorum sensing (QS) regulator influences the function of many target systems in *P. aeruginosa* including the QS regulators RhlR and MvfR (also known as PqsR; *Lee and Zhang, 2015*). We

tested the impact of the inactivation of RhlR or MvfR in a Δ*lasR* mutant background to understand how LasR influences community tolerance. We observed that recalcitrance of the Δ*lasR* mutant in the mixed community is dependent on the MvfR-PQS pathway, as the absence of a functional MvfR regulator resensitized the Δ*lasR* mutant to tobramycin (*Figure 3D*). On the other hand, the inactivation of RhlR QS regulator did not impact the sensitivity of the Δ*lasR* mutant (*Figure 3D*). None of the tested mutants, including Δ*lasR*, Δ*rhlR*, Δ*mvfR*, Δ*mvfR*Δ*lasR*, and Δ*rhlR*Δ*lasR*, resulted in a change in the endpoint CFUs of the other members in the mixed community in the presence or absence of tobramycin (*Figure 3—figure supplement 3*). Our data indicate that the tolerance of the Δ*lasR* mutant is biofilm-specific and implicates the MvfR-PQS QS system in this increased tolerance observed for this mutant in the mixed community.

## Phenazines induce *P. aeruginosa* tolerance in the mixed community

Studies examining the impact of interspecies interactions on LasR mutant variants have shown that phenazine production is stimulated in the presence of other microbial partners (*Hoffman et al., 2010*; *Cugini et al., 2010*). Furthermore, as the inactivation of the *mvfR* gene in a Δ*lasR* background rescued tobramycin sensitivity of *P. aeruginosa* in a polymicrobial environment (*Figure 3D*), and the MvfR-PQS system is crucial for phenazines production (*Déziel et al., 2005*; *Recinos et al., 2012*), we hypothesized that these molecules might be overproduced in the mixed community, thereby conferring tolerance to the Δ*lasR* mutant. That is, phenazines have previously been shown to drive tolerance of *P. aeruginosa* to various drug classes, including aminoglycosides (*Meirelles et al., 2021*; *Schiessl et al., 2019*; *Zhu et al., 2019*). For example, Schiessl and colleagues observed that inactivating phenazine production results in increased sensitivity of *P. aeruginosa* to tobramycin (*Schiessl et al., 2019*). Moreover, it has been reported that inactivation of the *lasR* gene drives overproduction of phenazines in *P. aeruginosa* (*Cabeen, 2014*), thus further suggesting a pivotal role of these redox-active molecules in *P. aeruginosa* tobramycin tolerance in our experimental conditions. We tested this hypothesis by quantifying the precursor molecule phenazine-1-carboxylic acid (PCA) produced by WT and associated mutants of *P. aeruginosa* grown as monospecies and mixed biofilm communities. Confirming our hypothesis, we measured significantly higher PCA levels in the mixed community containing the Δ*lasR* mutant when compared to WT *P. aeruginosa* (*Figure 4A*).

Also consistent with our hypothesis, blocking the production of phenazines in the Δ*lasR* background (ΔΔ*phz*Δ*lasR*, which lacks both *phz* operons) resensitized the Δ*lasR* mutant strain to tobramycin in the mixed community (*Figure 4B*). None of the tested mutants impacted the abundance of other members in the mixed community in the presence or absence of tobramycin (*Figure 4—figure supplement 1A*). We next sought to test the hypothesis that exogenous addition of phenazine induces tolerance of *P. aeruginosa* in the mixed community. Treatment of pre-formed biofilms with the phenazine pyocyanin triggered tolerance of WT *P. aeruginosa* in the mixed community in a dose-dependent manner (*Figure 4C*). The tested phenazine concentration did not impact endpoint CFUs of *P. aeruginosa* in all conditions tested (*Figure 4—figure supplement 1B*).

## Discussion

By leveraging culture-independent studies in combination with clinical metadata, we built a stable CF-relevant polymicrobial model system composed of *P. aeruginosa*, *S. aureus*, *S. sanguinis*, and *P. melaninogenica* (*Figure 1*). We used experimental conditions reflecting the nutritional (e.g. ASM) and environmental (e.g. anoxia) of the CF airway (*DePas et al., 2016*; *Worlitzsch et al., 2002*; *Palmer et al., 2005*). The community fell within the range of clinically observed communities in terms of species relative abundance (*Figure 1—figure supplement 4*), and furthermore, we observed several community-specific phenotypes (*Figure 1C*, *Figure 1—figure supplement 5*). In agreement with previous reports (*Filkins et al., 2015*; *Li et al., 2020*; *Scott et al., 2019*; *Stoner et al., 2022*), *S. aureus* and *Streptococcus* spp. showed decreased and increased growth, respectively, in the mixed community versus monoculture. While other studies have demonstrated that *P. aeruginosa*-secreted exoproducts can eradicate *S. aureus* in co-culture (*Filkins et al., 2015*; *Limoli et al., 2017*), we could maintain *S. aureus* viability in the mixed community for up to 14 days under the CF-like conditions used here (*Figure 1—figure supplement 6B*). Thus, the observations in our polymicrobial community mirror the capacity of these two pathogens to co-exist in the CF airway (*Camus et al., 2021*). Also,

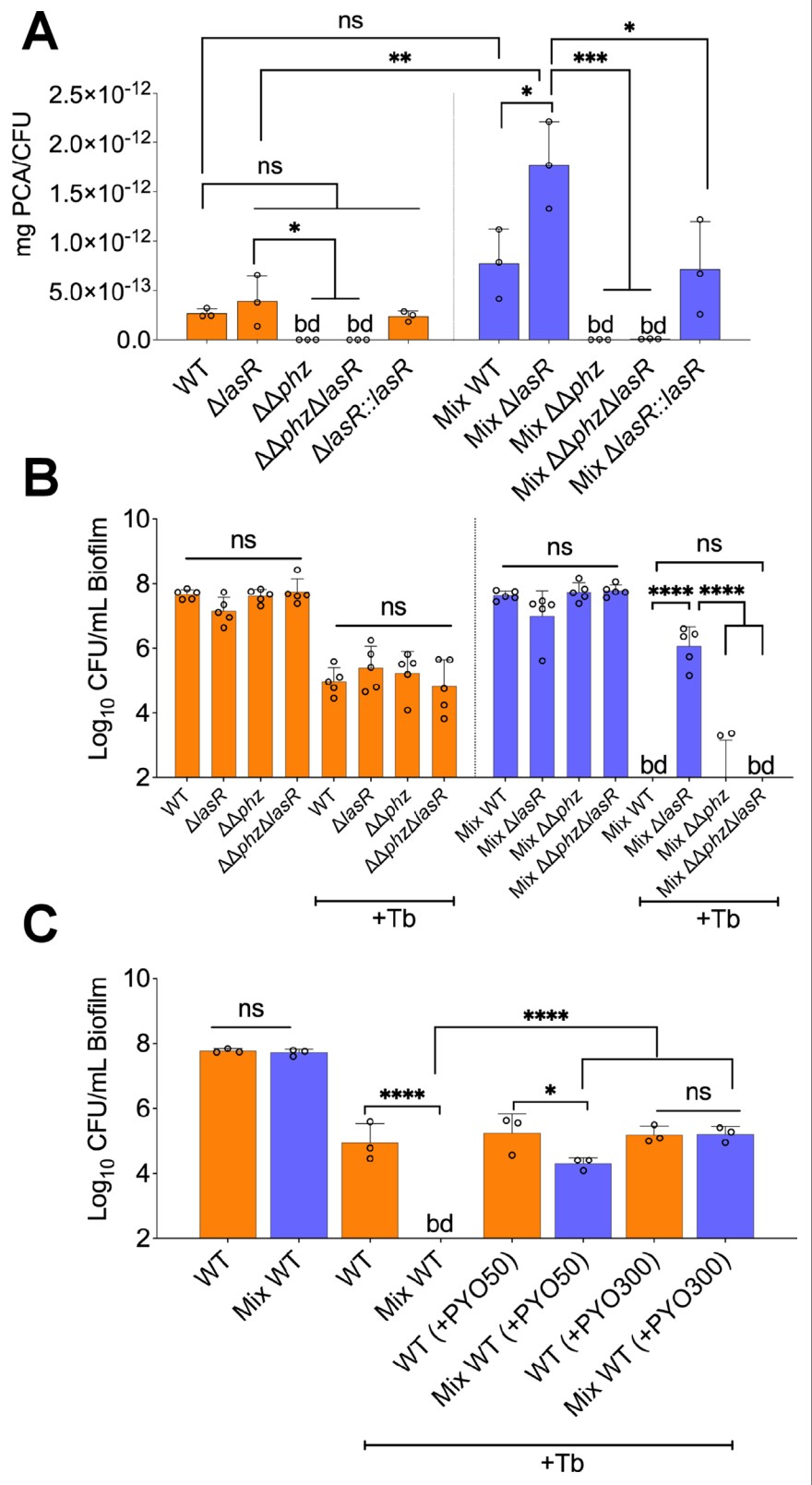

**Figure 4.** Phenazines drive tolerance of *P. aeruginosa* in mixed communities. (**A**) High performance liquid chromatography tandem mass spectrometry (HPLC-MS/MS) quantification of phenazines in monoculture and mixed communities (Mix) containing the indicated *P. aeruginosa* wild-type (WT) and mutant strains (n=3). (**B**) Colony forming units (CFUs) counts of *P. aeruginosa* (WT) and indicated mutants grown as monoculture or

*Figure 4 continued on next page*

*Figure 4 continued*

mixed communities (Mix; n=5). (**C**) Exogenous addition of phenazine to monoculture and mixed communities with WT *P. aeruginosa* treated with 100 µg/mL tobramycin (+Tb). Two physiologically relevant phenazine concentrations were tested; 50 µM (+PYO50) and 300 µM (+PYO300). Data for *S. aureus*, *S. sanguinis*, and *P. melaninogenica* counts are shown in *Figure 4—figure supplement 2*. Each data point presented in a column represents the average from at least three technical replicates performed at least on three different days (n=3). Statistical analysis was performed using ordinary one-way ANOVA and Tukey's multiple comparisons posttest with *=p<0.05; ***=p<0.001; ****=p<0.0001, ns = non-significant, and bd = below detection. Error bars represent SD.

The online version of this article includes the following figure supplement(s) for figure 4:

**Figure supplement 1.** Phenazine drive tolerance of *P. aeruginosa* in mixed communities: lack of impact on other community members.

**Figure supplement 2.** Impact of phenazine on community members treated with tobramycin: added phenazines and *P. aeruginosa* mutants do not impact other community members.

we ruled out the possibility of remaining oxygen in ASM negatively impacting the viability of *P. melaninogenica* by reproducing our results using an anoxic chamber (*Figure 1—figure supplement 2*). That is, we observed that *P. melaninogenica* can robustly grow as a planktonic or biofilm monospecies community in a medium capable of sustaining its growth (PGM), while this microbe fails to grow in ASM in monoculture (*Figure 1—figure supplement 2*). Thus, we argue that the mixed-community-specific growth of *Prevotella* spp. we observed across several conditions (*Figure 1C*, *Figure 1—figure supplement 5*, *Figure 2—figure supplement 6*) is not due to depletion of residual oxygen.

A major hurdle in the eradication of microbial communities detected in the CF airway is their resilience to front-line CF antimicrobials (*Orazi and O'Toole, 2019*; *Vandeplassche et al., 2019*). That is, while pathogens tested in monoculture demonstrate sensitivity to several antimicrobial classes (*Orazi and O'Toole, 2019*; *Vandeplassche et al., 2019*), these findings do not translate to improved outcomes in the clinic (*Heirali et al., 2020*; *Nelson et al., 2020*; *Heirali et al., 2017*). For example, a recent report shows that current testing methods fail to predict outcomes in pwCF (*Waters et al., 2019*). By modeling a polymicrobial community of six frequently encountered pathogens in the CF lung, Vandeplassche and colleagues compared the impact of several CF drugs on microbes grown as monocultures versus in a mixed community (*Vandeplassche et al., 2020*). Surprisingly, no shifts in sensitivities of any of the microorganisms were observed. These results could perhaps be explained by the differences in experimental conditions as Vandeplassche and colleagues used a rich medium, and incubations were performed under normoxic conditions (*Vandeplassche et al., 2020*). However, we acknowledge that the work of Vandeplassche and colleagues represents an important first step in tackling the impact of microbial interactions on antimicrobial susceptibility of CF pathogens.

Using more 'CF-like' experimental conditions informed by existing clinical data sets, we sought to test the hypothesis that growth in a mixed community would shift the sensitivity of CF microorganisms versus growth as monocultures. We focused on the front-line CF antimicrobial tobramycin as (i) this drug is the most heavily prescribed therapeutic to treat chronic CF lung infections (*Cystic Fibrosis Foundation Patient Registry, 2021*) but (ii) often does not eradicate pathogens (especially *P. aeruginosa*) from the airways of pwCF (*Heirali et al., 2020*; *Nelson et al., 2020*). Using our system, we made several observations. First, almost all tested *S. aureus* and *Streptococcus* spp. strains were protected from tobramycin-mediated killing in the mixed community versus monoculture (*Figure 2B*, *Figure 2C*, *Figure 2—figure supplement 4*, *Figure 2—figure supplement 5*). In a recently published study, Murray and colleagues observed that *S. aureus* can be protected from tobramycin eradication in the presence of *P. aeruginosa* (*Murray et al., 2022*), but this protection was abolished in the presence of a PQS operon inhibitor (*Murray et al., 2022*). Inactivation of the MvfR regulator, which controls the expression of the PQS operon (*Déziel et al., 2005*), did not sensitize *S. aureus* to tobramycin under our experimental conditions (*Figure 3—figure supplement 3*), indicating that other mechanisms might be driving this phenotype in our system. Furthermore, the observed increased tobramycin resilience of *Streptococcus* spp. in a polymicrobial environment (*Figure 2C*, *Figure 2—figure supplement 5*) is in agreement with a previous report (*Price et al., 2016*). Future studies will be necessary to identify the community-specific recalcitrance mechanisms employed by *S. aureus* and *Streptococcus* spp.

The decrease in the viability of *P. aeruginosa* in a tobramycin-treated community (*Figure 2A*, *Figure 2—figure supplement 1*, *Figure 2—figure supplement 2*, *Figure 2—figure supplement*

3, *Figure 2—figure supplement 7*) was unexpected. Recent culture-independent studies investigating the impact of tobramycin in pwCF do not show any appreciable decrease in *P. aeruginosa* after extended exposure to this antibiotic (*Heirali et al., 2020*; *Nelson et al., 2020*). These results prompted us to interrogate the molecular mechanisms driving this increased sensitivity of *P. aeruginosa* to tobramycin in the community. We observed biofilm-specific tolerance of a Δ*lasR* mutant grown in the mixed community (*Figure 3A*), a result repeated with two phenotypically LasR-defective CF clinical isolate variants (AMT0101-1-1 in *Figure 3C* and SMC1595 in *Figure 2—figure supplement 2*). Given that not all *P. aeruginosa* strains exhibited increased sensitivity in the mixed community, some strains may carry mutations in other pathways conferring tolerance to tobramycin in the mixed community. Indeed, clinical *P. aeruginosa* isolates can accumulate mutations including in the negative regulator MexZ, driving multidrug resilience (*Smith et al., 2006*).

We observed that phenazines mediate the community tolerance of the Δ*lasR* mutant and can induce tolerance of WT *P. aeruginosa* (*Figure 4B*, *Figure 4C*). Our rationale for testing the impact of these redox-active compounds in our system is supported by reports highlighting the capacity of these molecules to drive *P. aeruginosa* drug tolerance (*Meirelles et al., 2021*; *Schiessl et al., 2019*; *Zhu et al., 2019*). Also, strains lacking LasR function are known to overproduce phenazines (*Groleau et al., 2022*; *Mould et al., 2020*; *Cugini et al., 2010*; *Cabeen, 2014*; *Dekimpe and Déziel, 2009*). Interestingly, the highest levels of phenazine production by the Δ*lasR* mutant were observed in the mixed community (*Figure 4A*). PCA production of WT *P. aeruginosa* grown in the mixed community was not statistically significantly different from the concentration detected in monospecies biofilm, while the Δ*lasR* mutant did show a significant, twofold increase in PCA production in the community compared to the monoculture (*Figure 4A*). The mechanistic basis for the difference between the WT and Δ*lasR* mutant production of PCA in the mixed community is the subject of ongoing work. Finally, while we do see a role of phenazines in conferring tolerance, we acknowledge that other LasR-regulated factors may also contribute to this phenotype.

While the approaches used here have allowed us to explore community function, we acknowledge certain limitations to this study. First, while we provide a rationale for the selection of *P. aeruginosa*, *S. aureus*, *S. sanguinis*, and *P. melaninogenica* for use in our mixed community system, it is fair to ask, 'is four enough?' While the four microorganisms we have selected are a reasonable start, the model could be further improved by including additional CF-relevant species. However, adding pathogens such as *Burkholderia* or *Achromobacter* does not markedly increase the number of pwCF captured by our model (*Figure 1—figure supplement 1B and C*). Furthermore, while the in vitro polymicrobial model does not perfectly match the Pa.M1 and Pa.M2 mixed community compositions detected in the CF airway (*Hampton et al., 2021*), the abundances of *P. aeruginosa*, *S. aureus*, *S. sanguinis*, and *P. melaninogenica* all fall within the clinical range detected in these communities (*Figure 1—figure supplement 4*). More importantly, the increased sensitivity of *P. aeruginosa* to tobramycin treatment in the mixed community could be observed for *P. aeruginosa* co-cultivated over a 1000-fold range of the other microbial partners (*Figure 2—figure supplement 1*). These in vitro observations confirm the importance of studying microbes in the context of mixed-species communities but also suggest that the ratio of microbial partners relative to the *P. aeruginosa* population can vary with similar observed phenotypes.

Finally, we argue that this novel in vitro polymicrobial system will serve as a resource for the research community to probe novel community functions ranging from persistence to virulence and also for understanding the impact of host factors such as immune cells and host-derived metabolites. As such, all strains and protocols will be made freely available upon request, and we are happy to assist other groups in the implementation of this system.

## Materials and methods
### Bacterial strains and culture conditions

All the *P. aeruginosa*, *S. aureus*, *Streptococcus* spp., and *Prevotella* spp. strains used in this study are listed in *Supplementary file 2*. *P. aeruginosa* and *S. aureus* cultures were grown in Tryptic soy broth (TSB) with shaking at 37°C. *Prevotella* spp. cultures were grown in TSB supplemented with 0.5% yeast extract (YE), 5 μg/mL hemin, 2.85 mM L-cysteine hydrochloride, and 1 μg/mL menadione (PGM). *Streptococcus* spp. cultures were grown in Todd-Hewitt broth supplemented with 0.5%

YE (THY) at 37°C with 5% $CO_2$. Single colonies of each microbial species cultivated on either TSB solidified with 1.5% agar (TSA) for both *P. aeruginosa* and *S. aureus* species or TSA supplemented with 5% sheep's blood (blood agar) for *Streptococcus* spp. and *Prevotella* spp. were utilized to start overnight cultures in the abovementioned liquid media. ASM with 5 mg/mL of mucin (ASM) was prepared as previously described (*Turner et al., 2015*; *Palmer et al., 2007*) and supplemented with 100 mM 3-morpholinopropane-1-sulfonic acid (MOPS) to maintain a pH of 6.80 over the course of our studies.

## Microbial assays

Cultures experiments were performed using polystyrene flat-bottom 96-well plates. Cells from overnight liquid cultures of *P. aeruginosa*, *S. aureus*, *Streptococcus* spp., and *Prevotella* spp. were individually collected and washed twice (for *P. aeruginosa* and *S. aureus*) or once (for *Streptococcus* spp. and *Prevotella* spp.) in sterile PBS by centrifuging at 10,000 × *g* for 2 min. After the final wash, cells were resuspended in ASM with no mucin. The optical density ($OD_{600}$) was then measured for each bacterial suspension and diluted to an $OD_{600}$ of 0.2 in ASM. Monocultures and co-culture conditions were prepared from the $OD_{600}$=0.2 suspension and diluted to a final $OD_{600}$ of 0.01 for each microbial species in ASM corresponding to final bacterial concentrations of $1 \times 10^7$ CFU/mL, $3.5 \times 10^6$ CFU/mL, $1.2 \times 10^6$ CFU/mL, and $4.6 \times 10^6$ CFU/mL of *P. aeruginosa*, *S. aureus*, *Streptococcus* spp., and *Prevotella* spp. respectively. A volume of 100 µL of bacterial suspension all at a final $OD_{600}$ of 0.01 each in the mix was added to three wells. Plates were incubated using an AnaeroPak-Anaerobic container with a GasPak sachet (ThermoFisher) at 37°C for 24 hr. Then, unattached cells were aspirated with a multichannel pipette, and the pre-formed biofilms were replenished with 100 µL of fresh ASM on the bench and incubated for an additional 24 hr at 37°C using an AnaeroPak-Anaerobic container with a GasPak sachet (ThermoFisher). Similar experiments were performed using an anoxic environmental chamber (Whitley A55 - Don Whitley Scientific, Victoria Works, UK) with 10% $CO_2$, 10% $H_2$, and 80% $N_2$ mixed gas at 37°C, yielding results identical to those observed for the GasPak system. After a total of 48 hr of incubation, the planktonic fraction was collected, 10-fold serially diluted and plated on *Pseudomonas* isolation agar (PIA) for *P. aeruginosa* detection, Mannitol salt agar (MSA) for *S. aureus* quantification, TSB + 0.5% YE+1.5% agar supplemented with 5% sheep's blood, 10 µg/mL oxolinic acid, 10 µg/mL polymixin B for *Streptococcus* spp. (*Streptococcus* selective agar – SSA), TSB + 0.5% YE + 1.5% agar supplemented with 5% sheep's blood, 5 µg/mL hemin, 2.85 mM L-cysteine hydrochloride, 1 µg/mL menadione, 5 µg/mL vancomycin, and 100 µg/mL kanamycin for *Prevotella* spp. (*Prevotella* selective agar – PSA). For surface-attached communities, biofilms were washed twice with sterile PBS and resuspended in 50 µL sterile PBS. Bacteria in the biofilms were disrupted using a 96-pin metal replicator, and the resulting resuspended biofilm cells were 10-fold serially diluted in sterile PBS. Dilutions were plated on PIA, MSA, SSA, and PSA selective plates. *P. aeruginosa* and *S. aureus* selective plates were incubated for 16–20 hr at 37°C. *Streptococcus* spp. and *Prevotella* spp. selective plates were incubated in anoxic conditions for 24–48 hr at 37°C. After incubation, colonies were counted, and CFU/mL were determined for each microbial species. For the time course assays, CFU counts were performed for planktonic and biofilm fractions at 0, 3, 6, 12, 24, 48, 72, 96, 120, 144, 168, and 336 hr of monoculture and mixed communities grown in anoxic conditions. Pre-formed biofilms were replenished with fresh ASM every 24 hr for the duration of the experiment. For experiments with varying concentrations of *S. aureus*, *S. sanguinis*, and *P. melaninogenica* in monocultures and co-cultures, the organisms were grown from bacterial suspensions adjusted to an $OD_{600}$=0.8 in ASM. Suspensions were further diluted in ASM to an $OD_{600}$ of either 0.1, 0.001, 0.0001, or 0.00001 while maintaining *P. aeruginosa* at $OD_{600}$=0.01 (approximating $1 \times 10^7$ CFU/mL) in all conditions. The $OD_{600}$=0.1 dilution factor resulted in CFU/mL count average of $3.8 \times 10^8$ CFU/mL for *S. aureus*, $1.6 \times 10^8$ CFU/mL for *S. sanguinis*, and $1.0 \times 10^8$ CFU/mL for *P. melaninogenica*. The $OD_{600}$=0.001 dilution factor resulted in a CFU/mL count average of $6.7 \times 10^5$ CFU/mL for *S. aureus*, $1.1 \times 10^5$ CFU/mL for *S. sanguinis*, and $1.4 \times 10^5$ CFU/mL for *P. melaninogenica*. The $OD_{600}$=0.0001 dilution factor resulted in a CFU/mL count average of $4.2 \times 10^4$ CFU/mL for *S. aureus*, $3.3 \times 10^4$ CFU/mL for *S. sanguinis*, and $4.6 \times 10^4$ CFU/mL for *P. melaninogenica.* The $OD_{600}$=0.00001 dilution factor resulted in a CFU/mL count average of $5.6 \times 10^3$ CFU/mL for *S. aureus*, $4.4 \times 10^3$ CFU/mL for *S. sanguinis*, and $6.2 \times 10^3$ CFU/mL for *P. melaninogenica*. Data visualization and statistical analysis were done using GraphPad Prism 9 (version 9.2.0).

### Tobramycin susceptibility assay

Fresh stocks of tobramycin sulfate (Alfa Aesar) were prepared in sterile milli-Q water to a concentration of 50 mg/mL and subsequently diluted to a working stock of 5 mg/mL in ASM for each experiment. Triplicate wells containing pre-formed live cell biofilms of *P. aeruginosa*, *S. aureus*, *Streptococcus* spp., and *Prevotella* spp. grown as monoculture and mixed communities were initially grown for 24 hr as described in the 'Microbial assays' section. After incubation, the non-attached cells were collected, and pre-formed biofilms were exposed to 100 µg/mL tobramycin (100 µL/well) and further incubated for an additional 24 hr under anoxic conditions at 37°C. After a total of 48 hr of incubation, planktonic and biofilm cell fractions were sampled, 10-fold serially diluted, and plated as described previously. Data visualization and statistical analysis were done using GraphPad Prism 9 (version 9.2.0).

### MBC assay

Communities were grown in ASM for a total of 48 hr as described in the 'Microbial assays' section. The supernatants were collected then centrifuged for 10 min at 10,000 × $g$ and filtered-sterilized by using at 0.22 µM filter. To assess sterility of the supernatants, a volume of 5 µL for each condition was spotted on PIA, MSA, SSA, PSA, and blood agar plates. The supernatant spots were incubated at 37°C for PIA, MSA, and blood agar plates (aerobically) as well as SSA, PSA, and blood agar plates (anaerobically) for 24 hr. Serial twofold dilutions of tobramycin ranging from 500 µg/mL to 0.49 µg/mL were performed in either ASM or with the supernatants from *P. aeruginosa* monocultures and mixed communities. Duplicate wells of a sterile 96-well plate were used to inoculate 50 µL of various tobramycin concentration ranges in combination with either (i) 50 µL of sterile ASM (negative control) or (ii) 50 µL of the monoculture or mixed community supernatants. The plate was incubated at room temperature for a total of 24 hr. To compare the MBC concentrations of tobramycin after incubation in ASM or in supernatants, overnight cultures of *P. aeruginosa* PA14 were washed twice with sterile 1× PBS and resuspended in ASM to a final $OD_{600}$ of 0.1. 50 µL of the *P. aeruginosa* bacterial suspension were then transferred to two wells of the 96-well plate containing the various twofold dilutions of tobramycin made in either ASM or the culture supernatants. The plate was then incubated in aerobic conditions for 24 hr at 37°C. After the incubation, a 96-pin metal replicator was used to disrupt the cells formed at the bottom of the wells and then 3 µL of the resuspended cells were spotted on TSA plates and incubated at 37°C for 24 hr. Then, the MBC was determined by looking for absence of growth on the agar plate.

### Quantification of phenazines produced by *P. aeruginosa* by liquid chromatography tandem mass spectrometry

For phenazines quantification, communities of *P. aeruginosa* grown as monocultures and in mixed communities were prepared as described in the 'Microbial assays' section and inoculated in 12 wells (full row) of a 96-well plate. After incubation under anoxic conditions at 37°C, biofilms were disrupted into the planktonic fraction using at 96-well metal pin replicator, and cell suspension of 10 wells were transferred to a sterile 2.0 mL microtube. A volume of 1 mL of 100% dichloromethane (DCM - Thermo Scientific Chemicals) containing 0.02 ppm of 5,6,7,8-tetradeutero-4-hydroxy-2-heptylquinoline (HHQ-d4) as the internal standard (IS) was then added to the cells. The samples were then vortexed for 20 s and centrifuged at 16,000 × $g$ for 10 min at 4°C. The organic (bottom) phase of each sample was then transferred to a new 2.0 mL microtube. A second extraction of the remaining aqueous (top) phase was performed and added to the first organic extraction. The samples were evaporated at room temperature to a volume of 500 µL and then transferred to a conical high performance liquid chromatography (HPLC) glass vial before being completely evaporated. Samples were stored at –20°C until resuspended in 100 µL HPLC-grade acetonitrile and analyzed using liquid chromatography tandem mass spectrometry (LC-MS/MS) as previously described with minor modifications (*Lépine et al., 2018*). A 15 µL volume of sample was injected and analyzed by HPLC (Waters 2795; Waters, Mississauga, ON, Canada) using a Kinetex (100- by 3.0 mm) 5 µm EVO $C_{18}$ reverse-phase LC column (Phenomenex). The detector was a tandem quadrupole mass spectrometer (Quattro premier XE; Waters) equipped with a Z-spray interface using electrospray ionization in positive mode (ESI+). Nitrogen was used as a nebulizing and drying gas at flow rates of 15 and 100 mL/min, respectively. HPLC flow rate was 400 µL/min split to 40 µL/min by a Valco tee splitter. An acetonitrile-water gradient containing 1% acetic acid was used. In multiple reaction monitoring (MRM) mode, the following transition was monitored: 225 → 207

for PCA and 248→163 for the IS. The collision energies were set at 15 and 30 V, respectively, and the collision gas flow (argon) was set at 0.35 mL/min. Pure PCA (Sigma) was used as a standard for the method. Data visualization and statistical analysis were done using GraphPad Prism 9 (version 9.2.0).

### Quantification of 3-oxo-C$_{12}$-HSL produced by *P. aeruginosa*

Extraction of the LasR regulated signaling molecule 3-oxo-C$_{12}$-HSL from pure culture bacterial suspensions was performed as described in the 'Quantification of phenazines produced by *P. aeruginosa* by LC-MS/MS' section. Quantification of 3-oxo-C$_{12}$-HSL was done as previously reported by *Lépine et al., 2018*. Data visualization and statistical analysis were done using GraphPad Prism 9 (version 9.2.0).

### Tobramycin susceptibility assay in the presence of phenazines

Fresh stocks of tobramycin sulfate (Alfa Aesar) were prepared in sterile milli-Q water to a concentration of 50 mg/mL and subsequently diluted to a working stock of 5 mg/mL in ASM for each experiment. A 20 mM pyocyanin (Sigma) stock was prepared in 100% DMSO. Preparations containing either (i) ASM with or without vehicle controls, (ii) ASM supplemented with 100 µg/mL of tobramycin, (iii) ASM supplemented with 50 µM or 300 µM pyocyanin, and (iv) ASM supplemented with 100 µg/mL tobramycin and 50 µM or 300 µM pyocyanin were added to triplicate wells containing pre-formed biofilms of *P. aeruginosa*, *S. aureus*, *S. sanguinis*, and *P. melaninogenica* grown as monoculture and mixed community biofilms grown for 24 hr as described in the 'Microbial assays' section. Communities were incubated for 24 hr in anoxic conditions at 37°C. After a total of 48 hr of incubation, biofilm cell fractions were taken, 10-fold serially diluted, and plated as described previously. Data visualization and statistical analysis were done using GraphPad Prism 9 (version 9.2.0).

### Molecular techniques

In frame deletion of the *lasR* gene in a ΔΔ*phz* mutant was done using *Escherichia coli* SM10 $\lambda$ pir carrying the pEX18Gm-Δ*lasR* plasmid as previously reported (*Hogan et al., 2004*). Complementation of the *lasR* gene in a Δ*lasR* mutant background was done using *E. coli* S17 $\lambda$ pir carrying the knock-in pMQ30-*lasR* construct as previously described (*Clay et al., 2020*). Strains were confirmed by sequencing.

### 16S rRNA gene data analyses

The compendium of CF sputum microbiome data sets used here was developed from publicly available 16S rRNA reads as previously published (*Hampton et al., 2021*). Briefly, reads from sputum samples from 167 clinically stable subjects aged 8–69 from 14 CF centers in the United States and Europe were assigned to bacterial taxa at the genus level. Applying a gap statistic to unsupervised k-means clustering of Bray-Curtis distance of these 167 subject samples suggested that five community types best represented differences between the 167 subjects as previously described (*Hampton et al., 2021*). Here, we have used this compendium of taxonomically assigned reads from 167 pwCF to assess how many distinct genera, which ones, are required to capture 70% or more of taxonomically assigned 16S rRNA reads in each subject. See *Source code 1* for the script used to generate the data presented in *Figure 1A*, *Figure 1B*, *Figure 1—figure supplement 1*, and *Figure 1—figure supplement 4*.

### In silico community modeling

We used the SteadyCom method (*Chan et al., 2017*) to simulate the steady-state metabolism of a *P. aeruginosa*, *S. aureus*, *S. sanguinis*, and *P. melaninogenica* mixed community. SteadyCom performs community flux balance analysis by computing the maximal community growth while ensuring that all metabolites are properly balanced within each species and across the community. SteadyCom provides the capability to constrain the relative abundance of each species in the community to determine the main interspecies metabolic interactions. This simulation method is based on several simplifying assumptions, including spatial homogeneity and that all propagating species have the same growth rate at steady state. Outputs of each SteadyCom simulation included the community growth rate and species-dependent uptake and secretion rates of each extracellular metabolite. Model parameters, such as constraints in metabolite uptake rates, were taken from previous work modeling CF airway

communities (*Henson et al., 2019*). Four independent simulations, corresponding to the species abundances in the community types found to explain the variability in lung function, were performed. We calculated the relative abundance between *P. aeruginosa*, *S. sanguinis*, *P. melaninogenica*, and *S. aureus* within each community type, while ignoring remaining species. The simulations were then used to determine the main metabolites exchanged between the four species. See *Source code 2* for the SteadyCom script used to generate the modeling data presented in *Figure 1—figure supplement 1A*. Species abundance used for in silico modeling are presented in *Supplementary file 3*.

## Protease assays

Protease activity was assessed through casein degradation of tested *P. aeruginosa* strains grown on 1.5% TSA plates containing 1.5% sterile skim milk. Briefly, six colonies and appropriate controls grown on TSA plates were inoculated on milk plates and incubated overnight at 30°C for 24 and 48 hr. Casein degradation activity was observed by the presence or not of a zone of clearance around the inoculation site.

## 3-oxo-C$_{12}$-HSL autoinducer assays

The production 3-oxo-C$_{12}$-HSL of tested strain and associated controls was measured as described previously through the utilization of a 3-oxo-C$_{12}$-HSL-specific *lacZ* reporter (*Mould et al., 2022*). Briefly, the 3-oxo-C$_{12}$-HSL-specific *lacZ* reporter strain (DH161) was diluted to an OD$_{600}$=0.01 in lysis broth (LB) from an overnight culture. A volume of 100 µL was spread on LB agar plates supplemented with 150 µg/mL of 5-bromo-4-chloro-3-indolyl-β-D-galactopyranoside (X-gal) dissolved in 100% DMSO. Then, the plates were allowed to dry for 15 min in a biological safety cabinet. Once dry, a volume of 5 µL of OD$_{600}$ adjusted to 1.0 tested strains and controls (PA14 WT, Δ*lasR*, and Δ*lasI*Δ*rhlI*) was spotted on the pre-inoculated reporter strain. The plates were allowed to dry for an additional 15 min and then incubated at 37°C for 16–18 hr. Development of a blue halo around tested colonies was interpreted as a positive 3-oxo-C$_{12}$-HSL activity.

## Materials/data availability statement

All the necessary data needed for rigorous evaluation of the manuscript conclusions are accessible within the main text and/or associated supplementary files/figures/source data. Requests for reagents and other resources presented in the paper should be addressed to the corresponding authors.

## Acknowledgements

This work was supported by NIH grant R01 AI155424 to GAO, Canadian Institutes of Health Research (CIHR) operating grant MOP-142466 to ED, NIH/NIGMS 5 P20 GM130454 project support to DS, Cystic Fibrosis Foundation (CFF) grant HOGAN19G0 to DAH and CFF Postdoctoral Fellowship JEAN21F0 to FJP. Additional support is provided by the Cystic Fibrosis Foundation Research Development Program (STANTO19R0) and NIH P30-DK117469 (Dartmouth Cystic Fibrosis Research Center). We thank Dallas L Mould for technical help and helpful discussions. We also thank Dr. Benjamin D Ross for access to his anoxic environmental chamber and Dr. Sophie Robitaille, Dr. Courtney E Price, and Alexis R Ramsey for helpful discussions.

## Additional information

### Funding

| Funder | Grant reference number | Author |
|---|---|---|
| Cystic Fibrosis Foundation | JEAN21F0 | Fabrice Jean-Pierre |
| National Institutes of Health | R01 AI155424 | George A O'Toole |
| Canadian Institutes of Health Research | MOP-142466 | Eric Déziel |

| Funder | Grant reference number | Author |
| --- | --- | --- |
| National Institutes of Health | 5 P20 GM130454 | Daniel Schultz |
| Cystic Fibrosis Foundation | HOGAN19G0 | Deborah A Hogan |
| NIH | P30-DK117469 | Deborah A Hogan |
| Cystic Fibrosis Foundation Research Development Program | STANTO19R0 | Deborah A Hogan |

The funders had no role in study design, data collection and interpretation, or the decision to submit the work for publication.

### Author contributions

Fabrice Jean-Pierre, Conceptualization, Data curation, Formal analysis, Supervision, Funding acquisition, Validation, Investigation, Visualization, Methodology, Writing – original draft, Writing – review and editing; Thomas H Hampton, Data curation, Formal analysis, Validation, Visualization, Writing – review and editing; Daniel Schultz, Conceptualization, Formal analysis, Visualization, Methodology, Writing – review and editing; Deborah A Hogan, Resources, Validation, Writing – review and editing; Marie-Christine Groleau, Resources, Data curation, Formal analysis, Validation, Visualization, Methodology, Writing – review and editing; Eric Déziel, Resources, Validation, Methodology, Writing – review and editing; George A O'Toole, Conceptualization, Formal analysis, Supervision, Funding acquisition, Validation, Project administration, Writing – review and editing

### Author ORCIDs

Fabrice Jean-Pierre http://orcid.org/0000-0002-3056-8356
George A O'Toole http://orcid.org/0000-0002-2861-4392

### Decision letter and Author response

Decision letter https://doi.org/10.7554/eLife.81604.sa1
Author response https://doi.org/10.7554/eLife.81604.sa2

## Additional files

### Supplementary files

- Supplementary file 1. Minimal bactericidal concentration (MBC) of *P. aeruginosa* PA14 planktonic and biofilm cells treated with tobramycin exposed to the following conditions for 24 hr.

- Supplementary file 2. Strains and plasmids used in the study.

- Supplementary file 3. Species abundances in community types used for in silico modeling of metabolic flux.

- MDAR checklist

- Source code 1. 'R' script code used (i) to determine the abundance and prevalence data presented in *Figure 1A*, (ii) to determine the number of persons with cystic fibrosis (pwCF) for which ≥70% of 16S rRNA gene reads were assigned to *Pseudomonas*, *Staphylococcus*, *Streptococcus*, and *Prevotella* (*Figure 1B*) and *Pseudomonas*, *Staphylococcus*, *Streptococcus*, and *Prevotella* with *Burkholderia* or *Achromobacter* (*Figure 1—figure supplement 1B, C*), and (iii) to determine the microbial composition range of in vivo cystic fibrosis (CF) mixed communities compared with the in vitro model (*Figure 1—figure supplement 4A*). The 16S rRNA raw data published in *Hampton et al., 2021* was used with Source Code 1.

- Source code 2. SteadyCom (*Chan et al., 2017*) script used to determine predicted metabolic fluxes between *P. aeruginosa*, *S. sanguinis*, *S. aureus*, and *P. melaninogenica* for the *P. aeruginosa*-dominated (Pa.D), *Streptococcus*-dominated (Strep.D), and the Pa.M1/M2 mixed communities using the 16S rRNA relative abundances available in *Hampton et al., 2021*. Code used to generate *Figure 1—figure supplement 1A*.

## Data availability

Figure 1 - figure supplement 3 & Source Data 1 contains the numerical data used to generate the figure. Figure 1 - figure supplement 4 & Source Data 2 contains numerical data used to generate the figure. Source Code 1 contains the script used to generate Figure 1, Figure 1 - figure supplement 1 and Figure 1 - figure supplement 4. Source Code 2 contains the script used to generate the modeling data presented Figure 1 - figure supplement 1.

The following previously published dataset was used:

| Author(s) | Year | Dataset title | Dataset URL | Database and Identifier |
|---|---|---|---|---|
| Hampton TH, Thomas D, van der Gast C, O'Toole GA, Stanton BA | 2021 | Mild Cystic Fibrosis Lung Disease Is Associated with Bacterial Community Stability | https://www.ncbi.nlm.nih.gov/bioproject/?term=PRJEB30646 | NCBI BioProject, PRJEB30646 |

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
