## [Editor Report]

This article establishes a new model of the multispecies microbiome of infections of the cystic fibrosis airway and demonstrates its utility and reproducibility for testing the effectiveness of antibiotics and understanding how different bacteria interact. It shows that antibiotic susceptibility of the dominant pathogen *P. aeruginosa* changes in this mixed community relative to when grown alone, and indicates that a commonly-arising mutant of *P. aeruginosa* alters this sensitivity. This report is significant as a community resource and for its discovery of one reason why antimicrobial treatments can fail in polymicrobial infections.

---

## [Decision Letter]

**Decision letter after peer review:**

Thank you for submitting your article "Community composition shapes microbial-specific phenotypes in a cystic fibrosis polymicrobial model system" for consideration by *eLife*. Your article has been reviewed by 3 peer reviewers, one of whom is a member of our Board of Reviewing Editors, and the evaluation has been overseen by Arturo Casadevall as the Senior Editor. The following individuals involved in the review of your submission have agreed to reveal their identity: William DePas (Reviewer #2); Robert Quinn (Reviewer #3).

Essential revisions:

1) Further justification is required for why the authors jump to phenazine production as the singular lasR-associated phenotype they pursue. Details of phenazine experiments should be expanded.

2) There is a missed opportunity to analyze how interactions between species influence the productivity of each species and the greater community, with methods provided by one reviewer.

3) The methods for oxygen exposure during experiments and their effects require more clarity, particularly focusing on how it affects antibiotic tolerance and Prevotella growth in monoculture

4) The diluted mixed community experiment should be repeated to ensure reproducibility

*Reviewer #1 (Recommendations for the authors):*

L174-187 The difference in yield between biofilm and planktonic cultures is interesting. More interesting, and not explored, is how cultivation methods change the interactions between these species. With measures of growth yield from each of the species in monoculture at a given time point, and estimates of the numbers of each species in the inoculum, you can calculate how interactions between species influence the productivity of each species and the community as a whole. These calculations are important for a broader understanding of the community ecology of the model system and provide a baseline for further manipulations in the paper, eg following TOB treatment. This approach is detailed in Poltak and Cooper ISMEJ 2011 and is based on Loreau and Hector Nature 2002. I encourage the authors to analyze data using these equations to report realized productivity in mixture – who are the winners and losers of mixture, and how do these changes depend on cultural context? The way the data are currently presented in the figures makes these inferences nearly impossible.

TOB sensitivity in monoculture vs mixed culture: the reported changes are interesting and the results that lasR mutants influence susceptibility reasonable. However, the methods don't provide sufficient context for the reported findings and leave me wondering about intermediate ratios of strains, whether these changes require cells or merely supernatant, etc. This is fundamentally a methods development paper and these specific methods deserve further attention.

Re: lasR, please describe some other plausible reasons why lasR could have influenced TOB sensitivity other than phenazines. This section immediately jumps to these compounds as the cause of altered sensitivity and considers no other possible causes, which is problematic for such a pleiotropic regulator.

Reproducibility and sharing: cultivation and testing of this CF-relevant microbial community are presumably designed to be a resource for the community. Thus, a short statement in the main text describing the potential and limitations for sharing these resources would be a strength, beyond the standard availability statements in the supplementary material.

*Reviewer #2 (Recommendations for the authors):*

The paper is extremely interesting and, in my opinion, very valuable for infectious disease microbiologists and likely the broader microbiology field. My major concerns are related to the weaknesses identified in the public review.

1) How do your findings that phenazines can be protective against tobramycin tie in to your original observation that WT *P. aeruginosa* becomes susceptible to tobramycin in mixed communities? Figure 4 seems to indicate that WT *P. aeruginosa* makes more PCA in the mixed community than in a monospecies biofilm. Shouldn't that increase confer increased protection against tobramycin? Is that difference not significant? Is increased susceptibility possibly due to a decrease in a different type of phenazine? Even if phenazine production is not decreased in WT in the mixed biofilm community compared to monospecies biofilms, that result should be explicitly discussed.

2) Are you exposing cells to oxygen when changing the medium and when exposing cells to tobramycin? If so, how would you expect that to impact tolerance?

*Reviewer #3 (Recommendations for the authors):*

– Figure 1 c really adds nothing to the study. It needs to be more detailed to be useful.

– The metabolic flux analysis is poorly explained. Why would one expect the abundance of the organism to affect the metabolic flux? Is this not more of a binary analysis? It is unclear why this was done. For example, the wording 'multiple similarities' and 'distinguish' is very confusing. Can this be quantified?

– The decision to use a single 'mixed community model' is not well supported. This modeling aspect of the manuscript suffers from clarity and reasoning behind the approach.

– In fact lines 132-148 set a much better rationale for choosing these organisms, they are abundant and prevalent seems to make enough sense to me.

– Figure 1 supplement 3 is confusing. There are many bar graphs shown for the clinical strains, but the figure legend doesn't explain well what is being shown. Can this be better explained? This is an important experiment because the use of laboratory strains was peculiar after all the rationale for choosing the bacteria from people with CF.

– Figure 1 supplement 2B. It would be useful to show the variation in the experiment instead of a single point. Looking at the source data, the replicates are quite reproducible. This is important information for the reader and those trying to work on similar experiments to know that it is quite reproducible.

– The use of anaerobic culture conditions is of interest, especially with all of the pre-preparation of the cultures described in the methods. Were the washing and resuspending of cells done in anaerobic conditions? Might this explain the lack of growth of Prevotella spp. It is not clear if these preparations were done anaerobically. If Prevotella cultures are prepared and cultured in ASM anaerobically does the microbe grow? Or is it missing nutrients in ASM?

– Why was treating *Pseudomonas* alone with tobramycin and finding a reduction in planktonic growth surprising? It is an antibiotic after all.

– A recent experiment with sputum cultures in ASM as model communities shows that outcomes of competition in CF microbiome were driven by pH (Ghuneim et al. ISMEJ. 2022). The authors mention that they added MOPS to their cultures, can they comment on how this might affect competition and antibiotic outcomes in light of this similar manuscript?

– Figure 2 supplement 1, the 1000x diluted cultures of other bacteria with *P. aeruginosa* exposed to tobramycin has a very large error bar. I believe three technical replicates were performed, but only two points can be seen. Can this be explained? Repeated?

– Line 247 again brings up the question of whether the inability to culture Prevotella is due to not preparing in robust anaerobic conditions. This should be tested and shown.

– The pre-incubation with microbial supernatants does not preclude microbial inactivation as responsible for the increased killing. What if an intracellular toby deactivating enzyme is responsible? One could use mass spec to measure the amount of Toby that remains after the experiment.

---

## [Author Response]

Essential revisions:1) Further justification is required for why the authors jump to phenazine production as the singular lasR-associated phenotype they pursue. Details of phenazine experiments should be expanded.

We appreciate the reviewers’ comments and have expanded our justifications as to why we tested phenazines as a LasR-regulated target to explain the *P. aeruginosa* tolerance phenotype.

We have modified the text at lines 362–379 to the following:

“Studies examining the impact of interspecies interactions on LasR mutant variants have shown that phenazine production is stimulated in the presence of other microbial partners (1, 2). Furthermore, as the inactivation of the *mvfR* gene in a Δ*lasR* background rescued tobramycin sensitivity of *P. aeruginosa* in a polymicrobial environment (Figure 3D), and the MvfR-PQS system is crucial for phenazines production (3, 4), we hypothesized that these molecules might be overproduced in the mixed community, thereby conferring tolerance to the Δ*lasR* mutant. That is, phenazines have previously been shown to drive tolerance of *P. aeruginosa* to various drug classes, including aminoglycosides (5-7). For example, Schiessl and colleagues observed that inactivating phenazine production results in increased sensitivity of *P. aeruginosa* to tobramycin (6). Moreover, it has been reported that inactivation of the *lasR* gene drives overproduction of phenazines in *P. aeruginosa* (8), thus further suggesting a pivotal role of these redox-active molecules in *P. aeruginosa* tobramycin tolerance in our experimental conditions. We tested this hypothesis by quantifying the precursor molecule phenazine-1-carboxylic acid (PCA) produced by wild-type and associated mutants of *P. aeruginosa* grown as monospecies and mixed biofilm communities. Confirming our hypothesis, we measured significantly higher PCA levels in the mixed community containing the Δ*lasR* mutant when compared to WT *P. aeruginosa* (Figure 4A).”

2) There is a missed opportunity to analyze how interactions between species influence the productivity of each species and the greater community, with methods provided by one reviewer.

We agree with the reviewer’s comment that this type of analysis would be a good addition to the manuscript. We have added the productivity analyses to the paper (Figure 1 —figure supplement 3) and added the following paragraph at lines 206–228:

“Calculating the theoretical productivity (or yield) of each microorganism grown as planktonic or biofilm monospecies or as mixed communities, as defined using a reported method (9), and comparing these values to the experimentally determined populations revealed several interesting findings (Figure 1 —figure supplement 3). First, the yield of monospecies biofilm populations of *P. aeruginosa* and *S. aureus* was higher than planktonically-grown cells, while the opposite was observed for *S. sanguinis* (Figure 1 —figure supplement 3A). Furthermore, the observed productivity in the mixed planktonic community (denoted as “Mix (P) – Observed”) was lower than the theoretical productivity (i.e. assuming additivity of the CFU/mL yield counts of each species in the community, denoted as “Mix (P) – Theoretical”), suggesting antagonism among the different species grown in these conditions (Figure 1 —figure supplement 3A). Interestingly, examining the yield of the observed mixed biofilm and planktonic populations (both denoted as “Mix (P/B) – Observed”) revealed that the productivity of the mixed biofilm is higher than that of the planktonic community (Figure 1 —figure supplement 3A). There was no statistically significant difference in the yield of observed and theoretical mixed biofilm communities (Figure 1 —figure supplement 3A). However, taking a closer look at the productivity of each species in the observed mixed community revealed that there is a trend wherein *P. aeruginosa* and *S. aureus* have lower productivity in the mixed biofilm versus monoculture (P = 0.67) (Figure 1 —figure supplement 3B). On the other hand, *S. sanguinis* showed a statistically significant increase in productivity in a mixed biofilm community compared to the monospecies biofilm (P = 0.03) (Figure 1 —figure supplement 3B). Overall, these data indicate a complex relationship among the microbes in the community, which varies depending on whether they are growing planktonically or in a biofilm.”

3) The methods for oxygen exposure during experiments and their effects require more clarity, particularly focusing on how it affects antibiotic tolerance and Prevotella growth in monoculture

We agree with the reviewers and have integrated additional figures and associated text in the manuscript confirming that anoxic conditions were used. That is, we used an anoxic environmental chamber (10% CO_2_, 10% H_2_, 80% N_2_) for all steps in the experiment (thus the cultures were never exposed to oxygen), confirming that: (i) *P. melaninogenica* does not grow in ASM in monoculture, while as a positive control, we show it does grow when using a rich medium that has been reported previously to support its growth (10) (Figure 1 —figure supplement 2) and (ii) our observed changes in tobramycin susceptibility can be reproduced using the anoxic environmental chamber (Figure 2 —figure supplement 7). Taken together, these data indicate that our methods presented in the original manuscript (GasPak + anaerobic jar system) can rigorously and reproducibly recapitulate findings obtained in a controlled anoxic environment. The modifications to the revised manuscript are as follows:

Lines 198–204: “The impact of residual oxygen negatively influencing the growth of *P. melaninogenica* in monoculture was ruled out by performing these experiments using an anoxic environmental chamber (Figure 1 —figure supplement 2). That is, we did not detect CFU counts for either planktonic or biofilm populations of *P. melaninogenica* when grown in ASM in the anaerobic chamber, but as a positive control, significant growth was detected when using a medium shown previously to support growth of this microbe (10) (*Prevotella* Growth Medium, or PGM) (Figure 1 —figure supplement 2).”

Lines 406–414: “Also, we ruled out the possibility of remaining oxygen in ASM negatively impacting the viability of *P. melaninogenica* by reproducing our results using an anoxic chamber (Figure 1 —figure supplement 2). That is, we observed that *P. melaninogenica* can robustly grow as a planktonic or biofilm monospecies community in a medium capable of sustaining its growth (PGM) while this microbe fails to grow in ASM (Figure 1 —figure supplement 2). Thus, we argue that the mixed-community-specific growth of *Prevotella* spp. we observed across several conditions (Figure 1C, Figure 1 —figure supplement 5, Figure 2 —figure supplement 6) is not due to residual oxygen.”

Lines 290–293: “Growing and replenishing the preformed biofilm communities with fresh ASM supplemented or not with tobramycin using an anoxic environmental chamber resulted in similar phenotypes for all tested microorganisms (Figure 2 —figure supplement 7), indicating that the use of the GasPak system provides a robust anoxic environment.”

4) The diluted mixed community experiment should be repeated to ensure reproducibility

The diluted mixed community experiment was repeated a fourth time, yielding the same statistical conclusions. An updated “Figure 2 —figure supplement 1” was added to the paper. The highest (1:1000 – labeled in red in Figure 2 —figure supplement 1) dilution still yielded high variation, which could perhaps be explained by low (i.e., ~10^3^ CFU/mL) inoculum for *S. aureus*, *S. sanguinis,* and *P. melaninogenica* used in these experiments—see updated (“Microbial assays” paragraph of the “Materials and methods” section). Thus, the variation at low inoculum is robust and reproducible. The Materials and methods section was also updated to clarify the CFU counts used for those experiments. We have added modifications to the text as follows to address this critique:

Lines 526–532: “The optical density (OD_600_) was then measured for each bacterial suspension and diluted to an OD_600_ of 0.2 in ASM. Monocultures and co-culture conditions were prepared from the OD_600_ = 0.2 suspension and diluted to a final OD_600_ of 0.01 for each microbial species in ASM corresponding to final bacterial concentrations of 1x10^7^ CFU/mL, 3.5x10^6^ CFU/mL, 1.2x10^6^ CFU/mL and 4.6x10^6^ CFU/mL of *P. aeruginosa*, *S. aureus*, *Streptococcus* spp. and *Prevotella* spp. respectively. A volume of 100 µl of bacterial suspension all at a final OD_600_ of 0.01 each in the mix was added to three wells.”

Lines 558–570: “For experiments with varying concentrations of *S. aureus*, *S. sanguinis* and *P. melaninogenica* in monocultures and co-cultures, the organisms were grown from bacterial suspensions adjusted to an OD_600_ = 0.8 in ASM. Suspensions were further diluted in ASM to an OD_600_ of either 0.1, 0.001, 0.0001 or 0.00001 while maintaining *P. aeruginosa* at OD_600_ = 0.01 (approximating 1x10^7^ CFU/mL) in all conditions. The OD_600_ = 0.1 dilution factor resulted in CFU/mL count average of 3.8x10^8^ CFU/mL for *S. aureus*, 1.6x10^8^ CFU/mL for *S. sanguinis* and 1.0x10^8^ CFU/mL for *P. melaninogenica*. The OD_600_ = 0.001 dilution factor resulted in a CFU/mL count average of 6.7x10^5^ CFU/mL for *S. aureus*, 1.1x10^5^ CFU/mL for *S. sanguinis* and 1.4x10^5^ CFU/mL for *P. melaninogenica*. The OD_600_ = 0.0001 dilution factor resulted in a CFU/mL count average of 4.2x10^4^ CFU/mL for *S. aureus*, 3.3x10^4^ CFU/mL for *S. sanguinis* and 4.6x10^4^ CFU/mL for *P. melaninogenica.* The OD_600_ = 0.00001 dilution factor resulted in a CFU/mL count average of 5.6x10^3^ CFU/mL for *S. aureus*, 4.4x10^3^ CFU/mL for *S. sanguinis* and 6.2x10^3^ CFU/mL for *P. melaninogenica*.”

Reviewer #1 (Recommendations for the authors):L174-187 The difference in yield between biofilm and planktonic cultures is interesting. More interesting, and not explored, is how cultivation methods change the interactions between these species. With measures of growth yield from each of the species in monoculture at a given time point, and estimates of the numbers of each species in the inoculum, you can calculate how interactions between species influence the productivity of each species and the community as a whole. These calculations are important for a broader understanding of the community ecology of the model system and provide a baseline for further manipulations in the paper, eg following TOB treatment. This approach is detailed in Poltak and Cooper ISMEJ 2011 and is based on Loreau and Hector Nature 2002. I encourage the authors to analyze data using these equations to report realized productivity in mixture – who are the winners and losers of mixture, and how do these changes depend on cultural context? The way the data are currently presented in the figures makes these inferences nearly impossible.

We thank the reviewer for this great suggestion. We have revised our manuscript and added new analyses including measures of productivity of each microbe grown as planktonic or biofilm monoculture or in a mixed community (see “Essential Revisions” section, Comment #2).

TOB sensitivity in monoculture vs mixed culture: the reported changes are interesting and the results that lasR mutants influence susceptibility reasonable. However, the methods don't provide sufficient context for the reported findings and leave me wondering about intermediate ratios of strains, whether these changes require cells or merely supernatant, etc. This is fundamentally a methods development paper and these specific methods deserve further attention.

We agree with the reviewer and have made modifications in the Materials and methods section accordingly. More specifically, we have added a more detailed explanation for the target CFU/mL counts associated with the OD_600_ measurements for co-culture experiments as follows in the revised manuscript:

Lines 526–532: “The optical density (OD_600_) was then measured for each bacterial suspension and diluted to an OD_600_ of 0.2 in ASM. Monocultures and co-culture conditions were prepared from the OD_600_ = 0.2 suspension and diluted to a final OD_600_ of 0.01 for each microbial species in ASM corresponding to final bacterial concentrations of 1x10^7^ CFU/mL, 3.5x10^6^ CFU/mL, 1.2x10^6^ CFU/mL and 4.6x10^6^ CFU/mL of *P. aeruginosa*, *S. aureus*, *Streptococcus* spp. and *Prevotella* spp. respectively. A volume of 100 µl of bacterial suspension all at a final OD_600_ of 0.01 each in the mix was added to three wells.”

Lines 558–570: “For experiments with varying concentrations of *S. aureus*, *S. sanguinis* and *P. melaninogenica* in monocultures and co-cultures, the organisms were grown from bacterial suspensions adjusted to an OD_600_ = 0.8 in ASM. Suspensions were further diluted in ASM to an OD_600_ of either 0.1, 0.001, 0.0001 or 0.00001 while maintaining *P. aeruginosa* at OD_600_ = 0.01 (approximating 1x10^7^ CFU/mL) in all conditions. The OD_600_ = 0.1 dilution factor resulted in CFU/mL count average of 3.8x10^8^ CFU/mL for *S. aureus*, 1.6x10^8^ CFU/mL for *S. sanguinis* and 1.0x10^8^ CFU/mL for *P. melaninogenica*. The OD_600_ = 0.001 dilution factor resulted in a CFU/mL count average of 6.7x10^5^ CFU/mL for *S. aureus*, 1.1x10^5^ CFU/mL for *S. sanguinis* and 1.4x10^5^ CFU/mL for *P. melaninogenica*. The OD_600_ = 0.0001 dilution factor resulted in a CFU/mL count average of 4.2x10^4^ CFU/mL for *S. aureus*, 3.3x10^4^ CFU/mL for *S. sanguinis* and 4.6x10^4^ CFU/mL for *P. melaninogenica.* The OD_600_ = 0.00001 dilution factor resulted in a CFU/mL count average of 5.6x10^3^ CFU/mL for *S. aureus*, 4.4x10^3^ CFU/mL for *S. sanguinis* and 6.2x10^3^ CFU/mL for *P. melaninogenica*.”

Re: lasR, please describe some other plausible reasons why lasR could have influenced TOB sensitivity other than phenazines. This section immediately jumps to these compounds as the cause of altered sensitivity and considers no other possible causes, which is problematic for such a pleiotropic regulator.

We thank the reviewer for this comment. To probe mechanisms associated with the community tolerance phenotype of the *lasR* mutant, we tested ~20 mutants coding for global regulators, including mutants in metabolism regulation genes (*relAspoT*, *crc*) and/or mutants in genes coding for factors that are known targets implicated in *P. aeruginosa* tolerance to antimicrobials (*pel,* rhamnolipids, efflux pumps, siderophores). None of the tested mutants except for the *lasR* mutant resulted in increased tolerance to tobramycin. The observation that the double *lasRmvfR* mutant resulted in loss of the increased tolerance in the *lasR* mutant, in addition to our review of the recent literature, prompted us to focus on phenazines, as explained on Lines 362–379 of the updated manuscript (see “Essential Revisions” section Comment #1). We do note in the Discussion (Lines 476–478) that other LasR-regulated factors could also contribute to this phenotype, but that is the subject of future studies.

Reproducibility and sharing: cultivation and testing of this CF-relevant microbial community are presumably designed to be a resource for the community. Thus, a short statement in the main text describing the potential and limitations for sharing these resources would be a strength, beyond the standard availability statements in the supplementary material.

We agree with the reviewer’s comment. As such, the final part of the discussion has been modified to include the following text at Lines 498–502:

“Finally, we argue that this novel in vitro polymicrobial system will serve as a resource for the research community to probe novel community functions ranging from persistence to virulence and also for understanding the impact of host factors such as immune cells and host-derived metabolites. As such, all strains and protocols will be made freely available upon request, and we are happy to assist other groups in the implementation of this method.”

Reviewer #2 (Recommendations for the authors):The paper is extremely interesting and, in my opinion, very valuable for infectious disease microbiologists and likely the broader microbiology field. My major concerns are related to the weaknesses identified in the public review.1) How do your findings that phenazines can be protective against tobramycin tie in to your original observation that WT P. aeruginosa becomes susceptible to tobramycin in mixed communities? Figure 4 seems to indicate that WT *P. aeruginosa* makes more PCA in the mixed community than in a monospecies biofilm. Shouldn't that increase confer increased protection against tobramycin? Is that difference not significant? Is increased susceptibility possibly due to a decrease in a different type of phenazine? Even if phenazine production is not decreased in WT in the mixed biofilm community compared to monospecies biofilms, that result should be explicitly discussed.

We thank the reviewer for this comment. While we cannot completely rule out that other factors might shift tobramycin susceptibility of *P. aeruginosa* in a mixed community, our genetic, quantification, and biochemical data strongly suggest that phenazines play a major role in this phenotype. That is, adding phenazine to the mixed community treated with tobramycin restores tolerance of WT *P. aeruginosa*. We have updated Figure 4A, which now shows the statistical analysis comparing PCA production in WT *P. aeruginosa* grown as a monospecies vs mixed community. No statistically significant differences were observed. However, as specified by the reviewer, we have not measured all the possible phenazines produced by *P. aeruginosa*, and so cannot rule out contributions of other phenazines; but PCA and other phenazines such as pyocyanin can induce similar tolerance phenotypes in *P. aeruginosa*, as reported (6), suggesting a similar role for these molecules, at least for antibiotic tolerance. We argue that future work, outside the scope of this publication, will be necessary to completely understand how phenazines impact community tolerance of *P. aeruginosa*.

To address this critique, the following text was added at Lines 471–476:

“PCA production of WT *P. aeruginosa* grown in the mixed community was not statistically significantly different from the concentration detected in monospecies biofilm, while the *lasR* mutant did show a significant, 2-fold increase in PCA production in the community compared to the monoculture (Figure 4A). The mechanistic basis for the difference between the WT and *lasR* mutant production of PCA in the mixed community is the subject of ongoing work.”

2) Are you exposing cells to oxygen when changing the medium and when exposing cells to tobramycin? If so, how would you expect that to impact tolerance?

When the communities are replenished with fresh medium or treated with tobramycin, they are briefly exposed to oxygen. However, as shown in Figure 2 —figure supplement 7, when we repeated the key assays using an anoxic environmental chamber, we observed that the phenotypes are robust and reproducible as when using GasPaks and anaerobic jars.

Clarifications in the “Materials and methods” section have also been made at Lines 533–540:

“Plates were incubated using an AnaeroPak-Anaerobic container with a GasPak sachet (ThermoFisher) at 37 °C for 24 hours. Then, unattached cells were aspirated with a multichannel pipette and the pre-formed biofilms replenished with 100 µl of fresh ASM on the bench and incubated for an additional 24 hours at 37 °C using an AnaeroPak-Anaerobic container with a GasPak sachet (ThermoFisher). Similar experiments were performed using an anoxic environmental chamber (Whitley A55 – Don Whitley Scientific, Victoria Works, UK) with 10% CO_2_, 10% H_2_, 80% N_2_ mixed gas at 37 °C, yielding results identical to those observed for the GasPak system.”

Reviewer #3 (Recommendations for the authors):– Figure 1 c really adds nothing to the study. It needs to be more detailed to be useful.

We agree with the reviewer. We have decided to remove Figure 1C from the manuscript.

– The metabolic flux analysis is poorly explained. Why would one expect the abundance of the organism to affect the metabolic flux? Is this not more of a binary analysis? It is unclear why this was done. For example, the wording 'multiple similarities' and 'distinguish' is very confusing. Can this be quantified?

Varying abundances of *P. aeruginosa*, *S. aureus*, *S. sanguinis,* and *P. melaninogenica* (for all of which genome-scale metabolic models are available) result in shifts in the predicted metabolites produced by the various community types presented in “Figure 1 —figure supplement 1A”; the metabolite flux is indeed weighted to reflect community composition. As we have also previously published (13), these changes in metabolic flux can be used to predict the abundance of microbes that can be detected in the CF lung. We had hypothesized that the Pa.M1 and Pa.M2 community types, while identified as different by 16S rRNA gene amplicon clustering approaches, could be “collapsed” into a single community type that is distinct from a *P. aeruginosa* dominated community or a *Streptococcus*-dominated one. We do agree with the reviewer that future work using metabolomics approaches would be of interest to validate the modeling predictions between the different community types that we identified.

– The decision to use a single 'mixed community model' is not well supported. This modeling aspect of the manuscript suffers from clarity and reasoning behind the approach.

We have revised our manuscript and have updated our rationale for why a single “mixed community model” was selected. At Lines 130–140 it now reads:

“As previously reported (13), we performed metabolic flux analyses of the various community types to identify the top predicted exchanged metabolites driving the abundances of *P. aeruginosa*, *S. aureus*, *Streptococcus sanguinis* and *Prevotella melaninogenica* by using their known relative 16S rRNA gene abundance detected in the Pa.M1/Pa.M2, Pa.D, and Strep.D communities (Figure 1 —figure supplement 1A). Using this approach, we noted multiple similarities between the predicted metabolic fluxes of Pa.M1 and Pa.M2 that distinguished these communities from the *Pseudomonas*- and *Streptococcus*-dominated communities, for example, the fluxes of ethanol, lactate and acetate (Figure 1 —figure supplement 1A). Furthermore, as the predicted metabolic fluxes between Pa.M1 and Pa.M2 were similar, we considered these two mixed communities as functionally similar and decided to focus on the development of a single “mixed” community model.”

– In fact lines 132-148 set a much better rationale for choosing these organisms, they are abundant and prevalent seems to make enough sense to me.

We sought to use multiple approaches, including clinical observations, 16S rRNA gene amplicon clustering, and metabolic modeling approaches, to strengthen the reasons why we developed this in vitro polymicrobial community. We hope that this updated manuscript will clarify our rationale.

– Figure 1 supplement 3 is confusing. There are many bar graphs shown for the clinical strains, but the figure legend doesn't explain well what is being shown. Can this be better explained? This is an important experiment because the use of laboratory strains was peculiar after all the rationale for choosing the bacteria from people with CF.

We sought to use multiple approaches, including clinical observations, 16S rRNA gene amplicon clustering, and metabolic modeling approaches, to strengthen the reasons why we developed this in vitro polymicrobial community. We hope that this updated manuscript will clarify our rationale.

– Figure 1 supplement 2B. It would be useful to show the variation in the experiment instead of a single point. Looking at the source data, the replicates are quite reproducible. This is important information for the reader and those trying to work on similar experiments to know that it is quite reproducible.

We agree with the reviewer and have made the appropriate modifications to the figure so that experimental variation is also represented. Please see new “Figure 1 – —figure supplement 4B” as well as the associated “Source Data File 2”.

– The use of anaerobic culture conditions is of interest, especially with all of the pre-preparation of the cultures described in the methods. Were the washing and resuspending of cells done in anaerobic conditions? Might this explain the lack of growth of Prevotella spp. It is not clear if these preparations were done anaerobically. If Prevotella cultures are prepared and cultured in ASM anaerobically does the microbe grow? Or is it missing nutrients in ASM?

As mentioned in the “Essential Revisions” section, Comment #3, new experiments were performed using completely anoxic conditions (anoxic environmental chamber) that yielded similar observations, ruling out residual oxygen as the basis for the lack of growth of *Prevotella*.

– Why was treating Pseudomonas alone with tobramycin and finding a reduction in planktonic growth surprising? It is an antibiotic after all.

We agree with the reviewer that reduction in CFU counts of *P. aeruginosa* planktonic communities was not surprising. However, the interesting phenotype was the increased sensitivity of *P. aeruginosa* biofilms cells in a mixed community versus monoculture.

– A recent experiment with sputum cultures in ASM as model communities shows that outcomes of competition in CF microbiome were driven by pH (Ghuneim et al. ISMEJ. 2022). The authors mention that they added MOPS to their cultures, can they comment on how this might affect competition and antibiotic outcomes in light of this similar manuscript?

While not shown in this manuscript, we decided to add MOPS to ASM because our initial experiments indicated that endpoint growth of *P. aeruginosa* and *P. melaninogenica* was negatively impacted by low pH of ASM. These observations were likely driven by *S. sanguinis,* as the pH measurements for monocultures of this species were similar to the one observed in the mixed community (pH = ~4-5). Therefore, the mixed community was dominated by *S. sanguinis,* which was not reflective of the mixed communities in vivo. Furthermore, our in vitro observations seem to be in line with the data reported by Ghuneim and colleagues, where they report increased total bacterial load (driven by *Pseudomonas* and *Staphylococcus*) as a function of increase in pH through antibiotic-mediated inhibition of fermenting anaerobes. We do believe that in future work it would be interesting to look at the impact of pH on antibiotic susceptibility of microbes in various community types.

– Figure 2 supplement 1, the 1000x diluted cultures of other bacteria with *P. aeruginosa* exposed to tobramycin has a very large error bar. I believe three technical replicates were performed, but only two points can be seen. Can this be explained? Repeated?

As mentioned above in the “Essential Revisions” section, Comment #4, we repeated this experiment a fourth time, with similar observations. The highest (1:1000 – labeled in red in Figure 2 —figure supplement 1) dilution still yielded high variation, which could perhaps be explained by the very sparse (i.e. ~10^3^ CFU/mL) cells for *S. aureus*, *S. sanguinis* and *P. melaninogenica*; see updated “Microbial assays” paragraph of the “Materials and methods section” leading to stochastic effects at those concentrations.

– Line 247 again brings up the question of whether the inability to culture Prevotella is due to not preparing in robust anaerobic conditions. This should be tested and shown.

As presented in the “Essential Revisions” section, Comment #3, we have performed new experiments in strict anoxic conditions (Figure 1 —figure supplement 2) and have observed similar results.

– The pre-incubation with microbial supernatants does not preclude microbial inactivation as responsible for the increased killing. What if an intracellular toby deactivating enzyme is responsible? One could use mass spec to measure the amount of Toby that remains after the experiment.

We thank the reviewer for this comment. While we did not measure the remaining tobramycin concentration at the end of the experiment, we argue here that since we are still seeing killing of microbes such as the *Streptococcus milleri* group (Figure 2 —figure supplement 5A) in the mixed community, this suggests that the antimicrobial is still present at high enough concentrations to mediate its bactericidal activity. Furthermore, other studies such as in (14) have also reported a minimal bactericidal activity of tobramycin against *P. aeruginosa* situated at approximately the same concentration as the one reported in our study (i.e. 64 ug/mL).

References

1. Cugini C, Morales DK, Hogan DA. 2010. *Candida albicans*-produced farnesol stimulates *Pseudomonas* quinolone signal production in LasR-defective *Pseudomonas aeruginosa* strains. Microbiology 156:3096-3107.

2. Hoffman LR, Richardson AR, Houston LS, Kulasekara HD, Martens-Habbena W, Klausen M, Burns JL, Stahl DA, Hassett DJ, Fang FC, Miller SI. 2010. Nutrient availability as a mechanism for selection of antibiotic tolerant *Pseudomonas aeruginosa* within the CF airway. PLoS Pathog 6:e1000712.

3. Déziel E, Gopalan S, Tampakaki AP, Lepine F, Padfield KE, Saucier M, Xiao G, Rahme LG. 2005. The contribution of MvfR to *Pseudomonas aeruginosa* pathogenesis and quorum sensing circuitry regulation: multiple quorum sensing-regulated genes are modulated without affecting *lasRI*, *rhlRI* or the production of *N*-acyl-L-homoserine lactones. Mol Microbiol 55:998-1014.

4. Recinos DA, Sekedat MD, Hernandez A, Cohen TS, Sakhtah H, Prince AS, Price-Whelan A, Dietrich LE. 2012. Redundant phenazine operons in *Pseudomonas aeruginosa* exhibit environment-dependent expression and differential roles in pathogenicity. Proc Natl Acad Sci U S A 109:19420-5.

5. Meirelles LA, Perry EK, Bergkessel M, Newman DK. 2021. Bacterial defenses against a natural antibiotic promote collateral resilience to clinical antibiotics. PLoS Biol 19:e3001093.

6. Schiessl KT, Hu F, Jo J, Nazia SZ, Wang B, Price-Whelan A, Min W, Dietrich LEP. 2019. Phenazine production promotes antibiotic tolerance and metabolic heterogeneity in *Pseudomonas aeruginosa* biofilms. Nat Commun 10:762.

7. Zhu K, Chen S, Sysoeva TA, You L. 2019. Universal antibiotic tolerance arising from antibiotic-triggered accumulation of pyocyanin in *Pseudomonas aeruginosa*. PLoS Biol 17:e3000573.

8. Cabeen MT. 2014. Stationary phase-specific virulence factor overproduction by a *lasR* mutant of *Pseudomonas aeruginosa*. PLoS One 9:e88743.

9. Poltak SR, Cooper VS. 2011. Ecological succession in long-term experimentally evolved biofilms produces synergistic communities. ISME J 5:369-78.

10. Jang EY, Kim M, Noh MH, Moon JH, Lee JY. 2016. in vitro effects of polyphosphate against *Prevotella intermedia* in planktonic phase and biofilm. Antimicrob Agents Chemother 60:818-26.

11. Heirali A, Thornton C, Acosta N, Somayaji R, Laforest Lapointe I, Storey D, Rabin H, Waddell B, Rossi L, Arrieta MC, Surette M, Parkins MD. 2020. Sputum microbiota in adults with CF associates with response to inhaled tobramycin. Thorax 75:1058-1064.

12. Nelson MT, Wolter DJ, Eng A, Weiss EJ, Vo AT, Brittnacher MJ, Hayden HS, Ravishankar S, Bautista G, Ratjen A, Blackledge M, McNamara S, Nay L, Majors C, Miller SI, Borenstein E, Simon RH, LiPuma JJ, Hoffman LR. 2020. Maintenance tobramycin primarily affects untargeted bacteria in the CF sputum microbiome. Thorax 75:780-790.

13. Henson MA, Orazi G, Phalak P, O'Toole GA. 2019. Metabolic modeling of cystic fibrosis airway communities predicts mechanisms of pathogen dominance. mSystems 4:e00026-19.

14. Taylor PK, Zhang L, Mah TF. 2019. Loss of the two-component system TctD-TctE in *Pseudomonas aeruginosa* affects biofilm formation and aminoglycoside susceptibility in response to citric acid. mSphere 4:e00102-19.